# Overcoming resolution attenuation during tilted cryo-EM data collection

Sriram Aiyer [1], Philip R. Baldwin[1,2], Shi Min Tan[3], Zelin Shan [1], Juntaek Oh [4,5], Atousa Mehrani[1], Marianne E. Bowman[6], Gordon Louie[6], Dario Oliveira Passos [1], Selena Đorđević-Marquardt[1], Mario Mietzsch[7], Joshua A. Hull[7], Shuichi Hoshika [8], Benjamin A. Barad[9], Danielle A. Grotjahn [9], Robert McKenna [7], Mavis Agbandje-McKenna[7,15], Steven A. Benner[8], Joseph A. P. Noel [6,10], Dong Wang [4,10,11], Yong Zi Tan [3,12,13] ✉ & Dmitry Lyumkis [1,9,14] ✉

Structural biology efforts using cryogenic electron microscopy are frequently stifled by specimens adopting "preferred orientations" on grids, leading to anisotropic map resolution and impeding structure determination. Tilting the specimen stage during data collection is a generalizable solution but has historically led to substantial resolution attenuation. Here, we develop updated data collection and image processing workflows and demonstrate, using multiple specimens, that resolution attenuation is negligible or significantly reduced across tilt angles. Reconstructions with and without the stage tilted as high as 60° are virtually indistinguishable. These strategies allowed the reconstruction to 3 Å resolution of a bacterial RNA polymerase with preferred orientation, containing an unnatural nucleotide for studying novel base pair recognition. Furthermore, we present a quantitative framework that allows cryo-EM practitioners to define an optimal tilt angle during data acquisition. These results reinforce the utility of employing stage tilt for data collection and provide quantitative metrics to obtain isotropic maps.

Single-particle cryogenic electron microscopy (cryo-EM) has become a mainstream tool for macromolecular structural biology. Cryo-EM reconstructions resolved to near-atomic resolution are now routine and have shed light on many important biological phenomena[1,2]. Hardware and software advances have collectively led to the advent of true atomic resolution, where individual atoms are observed and distinguished within reconstructed maps[3–6]. Moreover, modern cryo-EM workflows allow users to gain insight into macromolecular structural heterogeneity, for example to help identify binding factors or to infer dynamics[7]. These and other capabilities demonstrate the power and the potential of the technique to visualize macromolecular machines, to explain their function with atomic-level precision, and to describe biological processes in dynamic terms.

Sample preparation for cryo-EM imaging requires first the vitrification of purified macromolecules within a thin aqueous layer. Subsequently, 2D projection images of individual macromolecular particles are recorded using a transmission electron microscope equipped with an electron detector. Assuming a homogenous particle subset, the projections differ depending on the orientation of each particle within the ice layer with respect to the electron beam. The determination of accurate Eulerian (Euler) orientation angles describing each distinct 2D particle-projection is the main goal of 3D refinement[8]. Once these orientations are determined, the particle views can be reconstructed to produce a 3D map[9]. Ideally, the particles will be randomly oriented, hence leading to an equal and uniform spread of Euler angles, yielding an isotropic map. However, in practice, uniform orientation distributions are rarely observed,

and non-uniform projection orientation distributions, wherein one or several views of the object predominate, are ubiquitous within cryo-EM datasets[10].

Non-uniformity of projection distributions is caused by the manner in which specimens are vitrified on grids prior to imaging[11]. Current grid vitrification techniques generate a thin aqueous film of vitreous ice, in which protein particles are embedded. For holey grids, the two interfaces at the boundary of the aqueous film attract macromolecular objects and lead to them adhering to either or both interfaces and adopting "preferred orientations". This results in over- and underrepresented views of the object, i.e., the set of projection views in the dataset and the coverage of Fourier space within the reconstruction are non-uniform. In the simplest scenario, non-uniform angular projection coverage leads to certain views becoming overpopulated, and in such cases the reconstructed map will be characterized by directional resolution anisotropy[12], with the poorest resolution being parallel to the direction of the electron beam[13]. However, in extreme cases, regions of Fourier space may be entirely missing. Severe directional resolution anisotropy compromises the fidelity of features within the reconstructed map, leading to ambiguities in biological interpretations (with and/or without an atomic model), and potentially stifling structure elucidation efforts[14].

A principal readout for directional resolution anisotropy is the 3D Fourier shell correlation (FSC) volume, a variation of the conventional 1D FSC curve[15]. Restricting the conventional FSC to a cone provides a consistency measure along a defined viewing direction, and repeating this calculation for different directions generates a complete measure of directional resolution across Fourier space[16,17]. Combining 1D conical FSC curves obtained from different angular directions yields a real 3D array termed the 3D FSC, which affords a quantitative assessment of the directional dependence of resolution for the reconstructed object. In addition to affecting directional resolution, non-uniform angular coverage also affects global resolution[13]. In the presence of non-uniform angular coverage, there is a net increase in noise variance and a corresponding attenuation of the 1D FSC curve that is used to define global resolution[13]. This attenuation can be estimated by a geometrical factor called the Sampling Compensation Factor (SCF)[13,14], which derives its name from the effect of the geometry of the sampling (relative to a uniform projection distribution) on the spectral signal-to-noise ratio (SSNR), and by extension the FSC. As shown previously[13], the effect of the projection distribution completely decouples from other experimental issues of the collection. The SCF is defined to be the harmonic mean of the sampling, over a shell of Fourier space, divided by the usual mean, and therefore ranges from 0 to 1. It provides a quantitative metric for evaluating an orientation distribution. Using the standard model of image formation for cryo-EM (which is used in the literature for many things such as the FSC = 1/7 resolution criterion), we derived that the SCF governs how much the SSNR is decremented relative to a uniform distribution. The SCF for the uniform distribution evaluates to the maximum value, 1, whereas an orientation distribution defined by perfect side-like views, which is completely (albeit still inhomogeneously) sampled, has an SCF that evaluates to 0.81. Because the side-like case, characterized by an SCF of around 0.81, represents the minimal case wherein Fourier space is completely sampled, we use this value as the baseline for a good map throughout the work. When there are large inhomogeneities in the sampling, the SCF rapidly decrements towards zero. Cases characterized by top-like distributions, which typically arise from preferentially oriented protein samples that are prevalent among experimental cryo-EM datasets, are frequently attenuated by a factor of as much as ten, in comparison to the uniform distribution (i.e., their SCFs are approaching 0.1, and even lower). In practice, reconstructions from samples characterized by such orientation distributions are not only attenuated in global resolution, but also characterized by severe anisotropy. Collectively, non-uniform sampling will affect directional

resolution anisotropy, as measured by the 3D FSC, and will also attenuate global resolution, as measured by the SCF.

A simple and generalizable solution to address the effects of non-uniform sampling caused by preferred orientation is to tilt the stage during data collection[15]. Although there are numerous possible experimental procedures that will alter the behavior of the object within the vitreous ice layer, their success is specimen-dependent and not generalizable (reviewed in ref. 18). Tilting the stage offers the simplest generalizable solution to the orientation bias problem. Due to the geometric configuration of the preferentially oriented sample with respect to the electron beam, for each cluster of orientations, tilting the stage yields a cone of illumination angles with respect to the cluster centroid, with cone half angle given by the tilt[15]. Tilting thus spreads out the orientation distribution of the imaged object, which in turn improves Fourier sampling. The quantitative benefits of tilting include a more isotropic 3D FSC and an SCF value that is closer to unity. Numerous successful case studies have benefited from the stage-tilt approach, including samples as small as ~43 kDa[19], and as large as ~100 MDa[20,21].

Despite the simplicity and generalizability of employing stage tilt during data collection, there have been multiple drawbacks to routine implementation. First, due to doming effects[22], there is generally more beam-induced movement. Second, due to variations in the focus gradient, tilting can lead to larger errors during image processing, for example when estimating the contrast transfer function (CTF). An unavoidable drawback is the increase in ice thickness. These possible hindrances lead to resolution attenuation and have historically limited the implementation of tilted data collection strategies.

The main goal of this study is to evaluate the effects of tilting the stage on single-particle cryo-EM reconstructions and to define optimal strategies for data collection and image analysis. Using three samples differing in molecular weight and characterized by isotropic 3D FSCs, we show that the quality of reconstructions derived from data collected at high stage tilt angles rivals that of the same specimen collected without stage tilt. These results indicate that collecting single-particle data with a fixed specimen stage-tilt on the microscope can result in negligible resolution attenuation in many cases. We implement these strategies using a biological sample of bacterial RNA Polymerase (RNAP) that contains in its active site an unnatural base pair (UBP) that matches 6-amino-3-(1′-β-D-2′-deoxy ribofuranosyl)−5-nitro-(1H)-pyridin-2-one in the template with 2-amino-8-(β − D-2′-ribofuranosyl)-imidazo-[1,2a]−1,3,5-triazin-(8H)−4-one as the incoming triphosphate. Both are part of an expanded DNA "alphabet"[23]. The polymerase adopts exclusively a single orientation on cryo-EM grids, making it difficult for cryo-EM to study pair recognition in this enzyme, and to allow those studies to guide synthetic biology. We show that high-angle stage tilts, combined with our processing workflows, can overcome issues with resolution anisotropy, leading to a molecular model that faithfully recapitulates the atomic details of the sample and describes how the UBP is bound.

Finally, using these observations, we devise a schema to identify an optimal tilt angle for a set of orientation distributions for diverse scenarios that would be encountered by the experimentalist. The presented strategies are generalizable and will guide the decision-making process of the experimentalist upon initiating data collection to deal with pathologically oriented samples.

## Results

### Experimental rationale and selection of test samples

Resolution attenuation when tilting the specimen stage arises from at least three sources. These include increases in ice thickness, larger beam-induced movement, and poorer CTF estimation. Having thin ice helps with obtaining high-resolution reconstructions, because the amount of inelastically and multiply scattered electrons is minimized, and there is less chance for particle overlap, among other factors[24].

Thus, cryo-EM practitioners frequently search for the thinnest possible ice[25], and methods for measuring ice thickness have been developed to aid this process[26]. Beam-induced movement[27–29] increases at higher tilt angles[15], due to the doming effect[22,30], where electron flux causes the ice to move up and down like a drum. Because the doming effect occurs perpendicular to the plane of the grid, it becomes more pronounced during imaging when the stage is tilted. The spread of defocus values corresponding to the particles may also lead to larger errors during CTF estimation. Given these effects, one would expect any imaging experiment that employs stage tilt to suffer from substantial resolution attenuation, and indeed these issues have limited the implementation of stage tilt for single-particle data collections. However, it is clear that high-resolution reconstructions can be obtained from both small[19] and large[20,21] specimens. We thus asked whether and how different stage tilt angles affect high-resolution cryo-EM analysis.

We selected as test specimens several highly symmetric protein oligomers that would not suffer from directional resolution anisotropy under any conditions (i.e., the SCF is ~1). Here, the high symmetry is essential, because symmetrization of the reconstruction can naturally bring the SCF to near unity, even if the specimen adheres to the air/water interface in a single orientation within the ice. Thus, the resolution attenuation due to incomplete sampling[13] is negligible and decoupled from the resolution attenuation due to the detrimental effects of tilted-stage imaging. The symmetry must be at minimum tetrahedral, and more preferably octahedral or icosahedral; as shown previously, symmetrization of C-fold or D-fold rotationally symmetric objects may not lead to improvements in sampling[14], particularly with specimens for which the preferred orientation is coincident with the symmetry axis relative to the direction of the electron beam. Finally, we also selected samples with a range of molecular weights, because variations in particle size should approximately relate to variations in the signal-to-noise ratio of individual particles.

## Resolution attenuation when the stage is tilted is negligible for large specimens such as Adeno-associated virus serotype 2 capsid (AAV2)

The first sample is AAV2[31], a 3.9 MDa assembly with a diameter of 250 Å. The 60-fold icosahedral (T = 1) symmetry of the AAV2 leads to near complete isotropic reconstructions[31]. We collected movies using the Titan Krios and the K2 direct electron detector with stage α-angle set to 0°, 10°, 20°, 30°, 40°, 50° or 60° (Supplementary Table 1). Our initial processing strategy was presented previously[15], also described in the Methods section. For comparative analysis, we selected an equal number of 7000 particles across all tilt angles, with similar defocus spread. As expected, the resulting reconstructions showed a clear trend of decreasing resolution as the tilt angle is increased (Fig. 1a); at 0° stage tilt, the resolution is 2.3 Å, but the resolution drops to 2.8 Å at 30° stage tilt, and 4.0 Å at 60° stage tilt.

To improve the data processing workflow, we first tested whether recent strategies for CTF estimation can improve the resolution of reconstructions generated from datasets collected using stage tilt. GCTF, which we employed previously, was used as the benchmark[15,32]. Applying the patch-based CTF estimation method in cryoSPARC[33] led to significant improvements in resolution for the tilted data collections (Fig. 1b). Reconstructions from 60° stage tilt reached below 3.0 Å resolution, and reconstructions from 30° stage tilt reached 2.3 Å. Notably, the reconstruction from 0° stage tilt also improved by 0.1 Å to 2.2 Å. Next, we refined the CTF values against a 3D model using RELION. This processing step did not improve the resolutions at lower tilt angles (0° and 10°), but did improve the resolutions at higher tilt angles, e.g. from 2.9 Å to 2.5 Å for 60° (Fig. 1c). When combined, both patch-based and per-particle refinement improved the resolution of the 60° reconstruction from 4.0 Å to 2.5 Å, a cumulative 1.5 Å increase. These results indicate that accurate CTF estimations are crucial to improving map quality and resolution for tilted datasets.

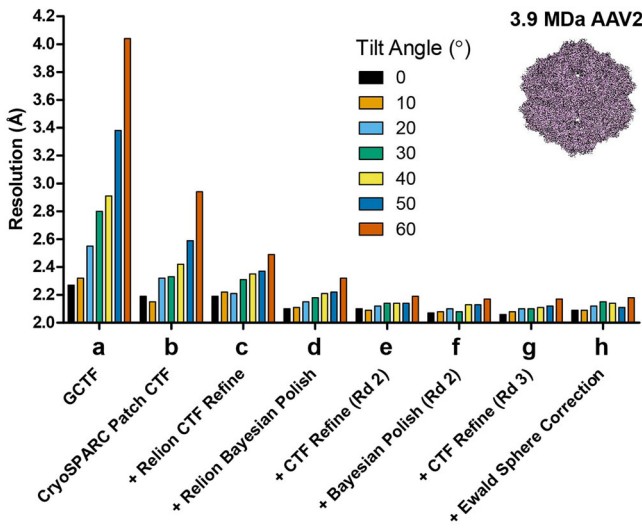

**Fig. 1 | Improvement in resolution for AAV2 at different tilt angles.** AAV2 cryo-EM data was collected at increasing stage tilt angles (0° black bars, 10° light orange bars, 20° light blue bars, 30° green bars, 40° yellow bars, 50° dark blue bars, 60° dark orange bars) and equalized for defocus, with 7000 particles used per tilt angle. The particles were processed using an identical workflow with global resolution at the end of each processing step plotted on the y-axis in Ångstroms (Å). **a** Baseline processing strategy presented in ref. 15. **b–h** Individual steps were added to the workflow, and improvements in resolution are plotted as lines. These include (**b**) patch-based CTF estimation in cryoSPARC, (**c**) CTF refinement in RELION, (**d**) Bayesian polishing in RELION, (**e**) a second round of CTF refinement and (**f**) Bayesian polishing, (**g**) a third round of CTF refinement, and (**h**) Ewald curvature correction.

We next sought to account for per-particle motion, as this will help restore information lost due to specimen movement during the imaging experiment[27–29,34]. In our original report[15], we used patch-based motion correction, but not per-particle motion correction, such as what is currently employed in RELION during Bayesian polishing[35]. The application of RELION Bayesian polishing improved the resolution of all reconstructions. For example, the 0° tilted reconstruction improved by 0.1 Å to 2.1 Å resolution, while the 60° tilted reconstruction improved to 2.3 Å resolution (Fig. 1d). Since RELION Bayesian polishing also applies an experimentally derived exposure weighting curve, the improvement in resolution may also arise from this effect.

For both the CTF refinement and Bayesian polishing, the 3D map is used as a reference. A better reconstruction should in turn yield more accurate CTF and particle movement parameters. Thus, we sought to determine if iterative CTF refinement and Bayesian polishing can improve the resolution of the tilted reconstructions. We found that the second round of CTF refinement and Bayesian polishing did not improve the 0° and 10° reconstructions (both remained at 2.1 Å), but it did improve reconstructions derived from data collected at higher tilt angles (Fig. 1e, f). No further improvements were observed with the third round of CTF refinement for reconstructions from data collected at 20°–60° stage tilt (Fig. 1g). Collectively, after several iterations, reconstructions from data collected at 20°–50° stage tilt reached identical resolution compared to reconstructions from data collected using 0° or 10° stage tilt. The resolution of the reconstruction from data collected at 60° stage tilt was just 0.1 Å less than the resolution of all other reconstructions. For completeness, we applied Ewald sphere curvature correction to the reconstructions[31], but observed no significant change in resolution (Fig. 1h). Higher order aberration corrections also did not improve the resolution, likely because the calculated beam tilt was minimal at under 0.1 mrad. These results show that, for large particles like AAV2, improved data processing strategies

can largely counter resolution attenuation caused by the imaging experiment when using stage tilt.

To validate the reconstructions, we compared the results from individual processing steps for data collected at 0° (Fig. 2 Left), 30° (Fig. 2 Middle), or 60° (Fig. 2 Right). First, we noted that the particles were not overly crowded in the raw images even at high-angle tilts, and high-resolution features were present in all sets of 2D class averages (Fig. 2a). As expected, the Euler angle distributions changed when the sample was tilted (Fig. 2b). However, the SCF remained ~1.0, and the 3D FSC was largely unchanged, irrespective of angular distribution and tilt angle, as expected from the icosahedral symmetry (Fig. 2c). Importantly, the global resolution estimate measured for the map-to-model FSCs matched the global resolution estimate measured for the half-map FSCs (Fig. 2d). Finally, the experimental reconstructions

showed clear side chain density and the presence of water molecules with the same level of detail (Fig. 2e). These observations confirm that the quality of the 2.2 Å resolution reconstruction derived from data collected using 60° stage tilt is nearly identical to that of the 2.1 Å resolution reconstruction derived from data collected using 0° stage tilt.

## Resolution attenuation at moderate tilt angle is negligible for medium-sized specimens such as apoferritin

Smaller proteins scatter fewer electrons and produce less signal within cryo-EM datasets. To determine if the trends for AAV2 hold for proteins of lower molecular weight, we repeated our experiments using the medium-sized protein, apoferritin[36], a 510 kDa multimeric assembly with octahedral symmetry and diameter of ~130 Å. Using a

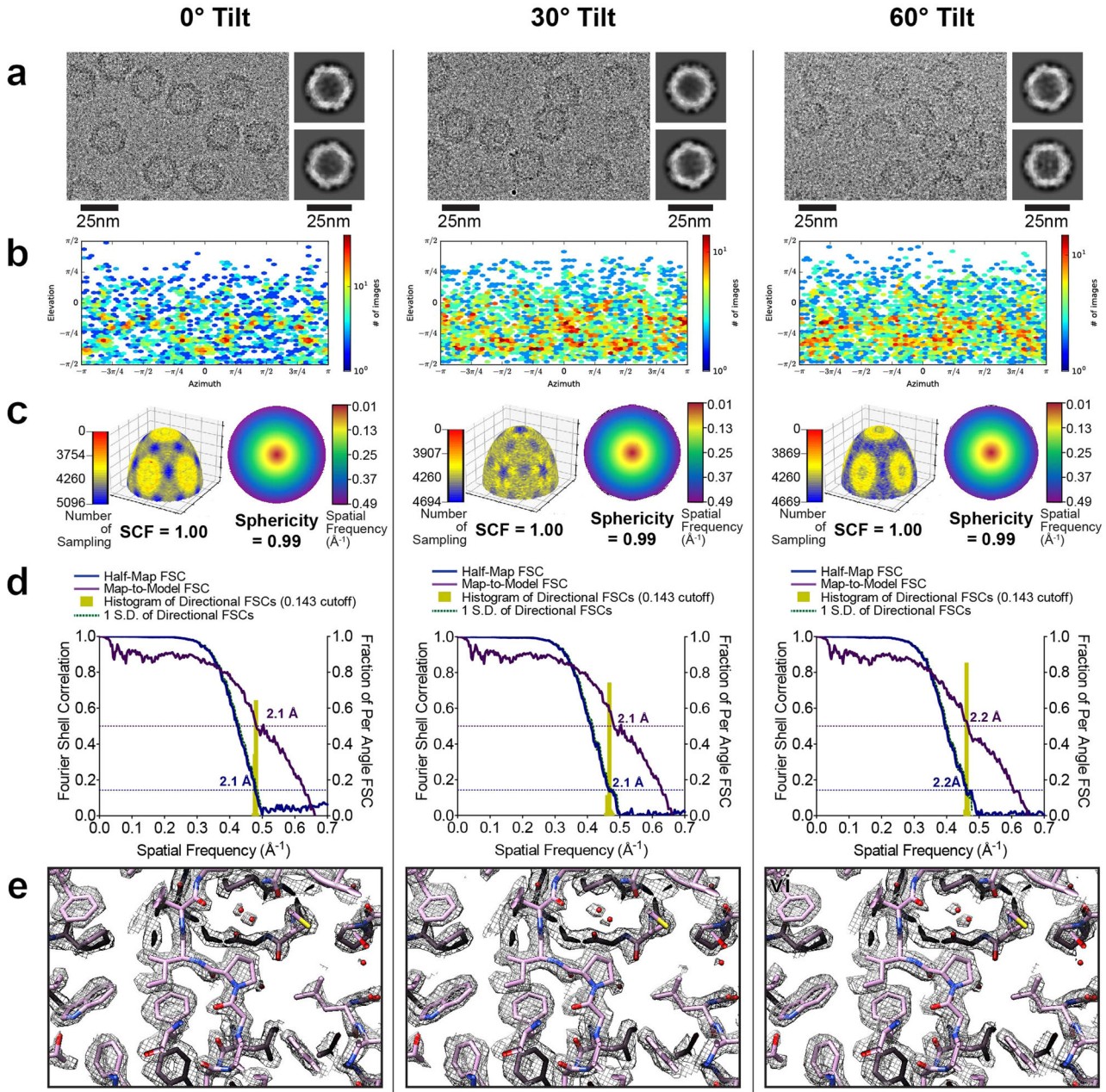

**Fig. 2 | Evaluation metrics for AAV2 at representative tilt angles.** Evaluation metrics of the cryo-EM data and reconstructed maps for the processing strategy outlined in Fig. 1. Results for untilted (left column), 30°-tilt (middle column) and 60°-tilt (right column) are compared. **a** Micrographs at 1.0 μm defocus, with corresponding 2D class averages at right. **b** Final Euler angle distribution. **c** Surface sampling plot with the sampling compensation factor (SCF) value below (left) and central slice through the 3DFSC, colored by resolution (right). **d** Half-map and map-to-model global FSC curves. **e** Map density with docked atomic model.

similar data collection and processing strategy as employed with AAV2 (Supplementary Table 2 and Methods), we found that our data processing workflow (patch CTF estimation, per-particle CTF refinement, and Bayesian polishing) significantly improved the resolutions of reconstructions across all tilt angles (Fig. 3): from 2.6 Å to 2.4 Å for 0° tilt data, from 3.1 Å to 2.5 Å for 30° tilt data, and from 3.8 Å to 2.7 Å for 60° tilt data (Fig. 3a–d). We observed further improvements in global resolution after the 2nd round of CTF refinement and Bayesian polishing (Fig. 3e, f), but not after the 3rd round (Fig. 3g). Finally, we noted small improvements when correcting for higher-order optical aberrations, yielding final resolutions of 2.2 Å, 2.3 Å, and 2.5 Å for reconstructions from 0°, 30°-tilted, and 60°-tilted data, respectively (Fig. 3h).

With these improvements, we found that the resolution attenuation at moderate tilt angles of 30° was negligible compared to 0°, with both these tilt angles for data collection producing maps at a nominal 2.2 Å resolution. For reconstructions from 0° (Fig. 4, left), 30°-tilted (Fig. 4, center), and 60°-tilted (Fig. 4, right) data, map features were observed in raw particles and 2D class averages (Fig. 4a). Although the angular distributions became increasingly dispersed (Fig. 4b), this dispersion was largely inconsequential to the sampling distribution or the 3D FSC (Fig. 4c). The derived experimental resolution was corroborated via similar half-map and map-to-model resolutions (Fig. 4d) and nearly identical looking density maps, with features that correspond to the expected nominal resolution (Fig. 4e).

In support of and consistent with the above findings, in early experimental attempts with apoferritin as a test specimen, we collected data with the stage α-angle set to 0° or 30°, i.e., omitting the other angles. After image processing, we were able to resolve maps to nearly identical 1.9 Å for reconstructions from both 0° data and 30°-tilted data (Supplementary Table 3 and Supplementary Fig. 1). The higher resolution for these maps, in comparison to the maps described above, can be attributed to the larger data size. In yet another set of

experiments, we employed the ~700 kDa 20S proteasome, which is characterized by D7 symmetry but has a non-uniform orientation distribution due to the side-like sampling of particle views. We collected cryo-EM data with the stage α-angle set to 0°, 10°, 20°, 30°, 40°, 50°, but omitted a dataset employing 60° stage tilt (Supplementary Table 4). After image processing, we observed similar trends as with apoferritin (Supplementary Figs. 2, 3).

Collectively, reconstructions from 0° data and as high as 50°-tilted data were largely similar. However, when the stage α-angle was increased to 60°, the resolution of the apoferritin reconstruction was ~0.3 Å poorer compared to the others. It is possible that workflow optimization may yield further resolution gains, especially with future improvements in hardware/software, but we were not able accomplish this using current strategies. The ancillary data in which the complete tilt series was not collected were fully supportive of our main findings. We conclude that resolution attenuation is largely negligible for medium-sized specimens, except for the highest tilt angles.

## Resolution is attenuated with increasing stage tilts for small specimens such as DNA protection during starvation protein (DPS)

We next turned our attention to our smallest specimen, the 230 kDa DPS[37], which is characterized by tetrahedral symmetry. We again collected datasets with the stage α-angle set to 0° or tilted up to 60°, in 10° increments (Supplementary Table 5). As with the larger protein specimens, we observed most of the improvements in resolution after one round of patch CTF estimation and Bayesian polishing (Fig. 5a–c). However, CTF refinement before and after Bayesian polishing did not improve the resolution, likely because of the low signal-to-noise ratio for the smaller DPS particles. We observed only minor resolution changes in the second round of Bayesian polishing and final CTF refinement (Fig. 5d, e), and no further changes afterwards. Correcting for higher-order aberrations yielded small gains (Fig. 5f). Although tilting the stage produced a slightly poorer reconstruction for DPS compared to the reconstruction when the stage was not tilted, our revised data-processing workflow again yielded significant improvements in resolution for reconstructions across all tilt angles—from 3.0 Å to 2.6 Å for 0°; from 3.6 Å to 3.0 Å for 30°; and from 4.4 Å to 3.3 Å for 60°. All the metrics for evaluating map quality were consistent with the reported nominal resolution values (Fig. 6a–d). Importantly, our results show that with appropriate data processing, tilting the stage can be a viable strategy to obtain high-resolution reconstructions even for relatively small proteins suffering from preferred orientation. This in turn will allow for accurate derivation of atomic models, as evidenced by the final quality of the experimental maps (Fig. 6e). As specimen preparation, hardware, and data-processing algorithms advance, we expect that resolution attenuation will diminish accordingly in the future.

## High-angle stage tilt addresses the orientation bias problem in RNAP and reveals how an unnatural base pair is recognized

We then used the *E. coli* RNAP to evaluate the generalizability of our tilted-stage image acquisition in a case of special interest to synthetic biology[38]. Here, we asked whether the improved strategies for processing cryo-EM data can help understand how this polymerase transcribes unnatural base pairs not found in the standard DNA or RNA "alphabets" – in particular, how Us bind, how mispairing is recognized, and how next generation UBPs might be designed. This is especially challenging for RNAP, which is known to exhibit strong preferential orientation on cryo-EM grids[39]; conventional data collection strategies have typically yielded maps characterized by severe resolution anisotropy characterized by elongated features. The inclusion of detergent can produce additional views[39], leading to resolvable features within cryo-EM maps, however, the resulting orientation distribution remains non-uniform. Further, it is well-known that the addition of detergent is

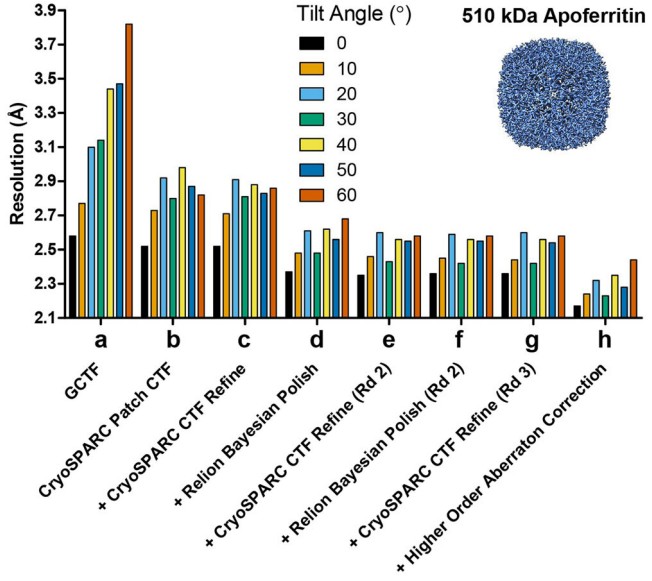

**Fig. 3 | Improvement in resolution for apoferritin at different tilt angles.** Apoferritin cryo-EM data was collected at increasing stage tilt angles (0° black bars, 10° light orange bars, 20° light blue bars, 30° green bars, 40° yellow bars, 50° dark blue bars, 60° dark orange bars) and equalized for defocus, with 17,000 particles used per tilt angle. This figure follows the format of Fig. 1. The particles were processed using an identical workflow with global resolution at the end of each processing step plotted on the y-axis in Ångstroms (Å). **a** Baseline processing strategy used in ref. 15. **b**–**h** Individual steps were added to the workflow and improvements in resolution are plotted as lines.

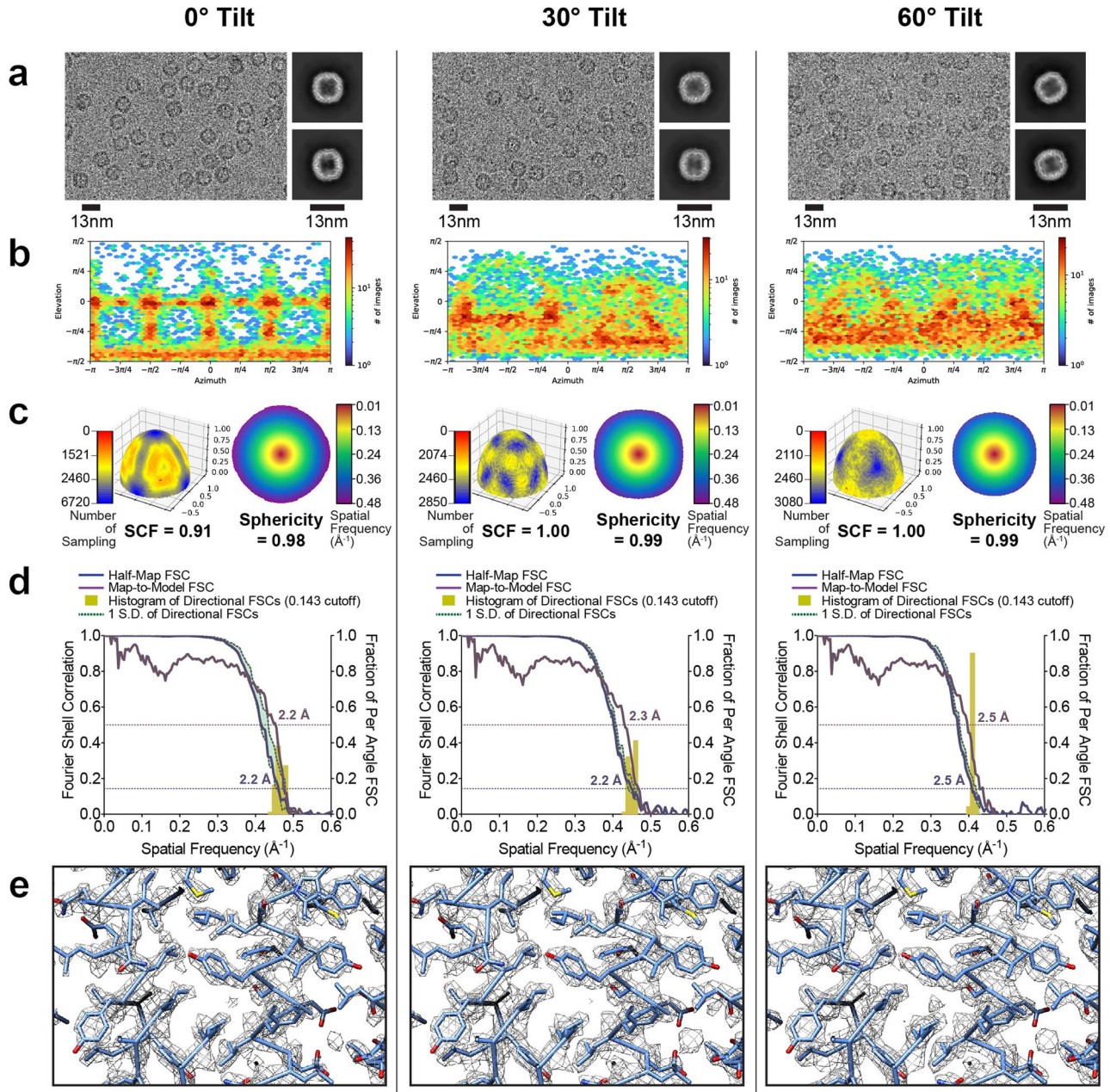

**Fig. 4 | Evaluation metrics for apoferritin at representative tilt angles.** Evaluation metrics of the cryo-EM data and reconstructed maps for the processing strategy outlined in Fig. 3. Results for untilted (left column), 30°-tilt (middle column) and 60°-tilt (right column) are compared. **a** Micrographs at 1.3 μm defocus, with corresponding 2D class averages at right. **b** Final Euler angle distribution. **c** Surface sampling plot with the sampling compensation factor (SCF) value below (left) and central slice through 3DFSC, colored by resolution (right). **d** Half-map and map-to-model global FSC curves. **e** Map density with docked atomic model.

not a generalizable strategy to address preferential orientation for other protein assemblies. The overall molecular weight is approximately 480 kDa, which is comparable to apoferritin. But RNAP lacks internal symmetry, which precludes mitigating the effects of directional resolution anisotropy.

To confirm prior observations, we collected several tilt series from a grid used for single-particle analysis. In the resulting tomograms, almost all particles reside at either one of the two air/water interfaces (Fig. 7 and Supplementary Movie 1) and thus are not randomly oriented within vitreous ice.

We began by collecting a dataset for RNAP with the stage α-angle set to 0° and obtained a map nominally resolved to ~3.7 Å (Supplementary Table 6). However, this map suffered from severe directional resolution anisotropy and was characterized by an SCF of ~0.4. To determine an optimal tilt angle for data collection, we used the sampling tool we developed that was previously extended to the community, which determines, based on the SCF of the reconstruction from untilted data, how the orientation distribution (and by extension, the resulting SCF) can improve when the stage is tilted[13,14]. We predicted that a 51° tilt-angle should yield an orientation distribution characterized by an SCF of ~0.81, which would result in a fully sampled reconstruction. From a dataset collected with the stage α-angle set to 51°, we obtained reconstructed maps from heterogeneous refinement that were characterized by a sampling distribution with a mean SCF of ~0.7 (Supplementary Table 6 and Supplemental Fig. 4) and substantially less anisotropy than the reconstructed map from the untilted data. The discrepancy between the predicted SCF of 0.81 and the experimentally measured SCF of 0.7 from data collected at the

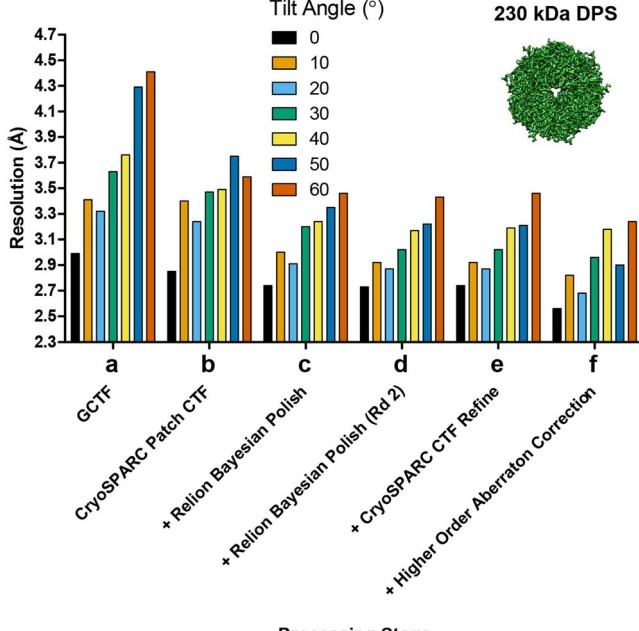

**Fig. 5 | Improvement in resolution for DPS at different tilt angles.** DPS cryo-EM data was collected at increasing stage tilt angles (0° black bars, 10° light orange bars, 20° light blue bars, 30° green bars, 40° yellow bars, 50° dark blue bars, 60° dark orange bars) and equalized for defocus, with 1750 particles used per tilt angle. This figure follows the format of Fig. 1. The particles were processed using an identical workflow with global resolution at the end of each processing step plotted on the y-axis in Ångstroms (Å). **a** Baseline processing strategy used in ref. 15. **b–f** Individual steps were added to the workflow and improvements in resolution are plotted as lines.

predicted optimal 51° tilt-angle can be attributed to angular mis-assignment of particles within the untilted dataset. The erroneous inclusion of a small amount of false-positive projections would yield overly optimistic SCF values for the untilted orientation distribution, which would in turn translate to an underestimate of the predicted optimal tilt angle, as further explained in the Discussion section. Thus, we increased the stage α-angle to 60°, which is close to the maximum amount of stage tilt that can be set on most electron microscopes, and collected a dataset (Supplementary Table 6). We calculated independent reconstructions from a subset of 72,000 particles with each of the 0° and 60°-tilted datasets. Analogous to the symmetric particles, we obtained iterative improvements in resolution with the updated processing approaches for reconstructions from 60°-tilted data (Supplementary Fig. 5). For reasons that are explained in Supplementary Note 1, the improvements in global resolution for RNAP reconstructions obtained from 0° data are not reliable and cannot be compared with the 60°-tilted data.

We next evaluated comparatively the RNAP cryo-EM maps using the standardized metrics. Raw particles could readily be observed for both 0° data and 60°-tilted data, but as expected, the 2D averages were consistent with distinct sets of views (Fig. 8a). Typical of a preferentially-oriented sample with C1 symmetry, for RNAP only one clear orientation was evident, with a narrow Euler angle distribution (Fig. 8b, left); when the stage was tilted, the distribution became conical with respect to the cluster centroid, with cone half angle given by the tilt (Fig. 8b, right). The reconstruction from untilted data hyper-sampled the region of Fourier space in and around a single plane and had a characteristically low SCF value (Fig. 8c, left), but the sampling improved when the stage was tilted, and the SCF was higher (Fig. 8c, right). As described previously[15], we observed a large discrepancy between the half-map and map-to-model FSCs for the reconstruction

from 0° data (Fig. 8d, left); the half-map and map-to-model FSCs were more similar for the reconstruction from 60°-tilted data (Fig. 8d, right). Most importantly, significant qualitative differences between the reconstructed maps were clearly evident. With the reconstruction from 0° data, the features of the map were severely elongated, and it was nearly impossible to fit, or properly interpret, an atomic model from the density (Fig. 9a top). In stark contrast, the reconstruction from 60°-tilted data produced a map with features fully consistent with the model (Fig. 9a bottom). This latter map was comparable in quality to typical reconstructions derived using cryo-EM[39,40] and readily lent itself to biological interpretations. Merging the untilted and tilted datasets did not yield an improvement in the quality of reconstructions compared to the tilted dataset alone (see Supplementary Note 2).

Based on the model derived from the density obtained from the 60°-tilted data alone, we are able to reveal important features of *E. coli* RNAP elongation complexes. These include base stacking of upstream RNA:DNA hybrid and side chains of conserved structural motifs, such as bridge helix (Fig. 9b). We observe clear undistorted density corresponding to how the incoming unnatural nucleotide substrate binds at the canonical addition site that base pairs with i + 1 template base (Fig. 9b), as well as the density attributed to trigger loop motif.

## Geometric and sampling considerations for defining an optimal tilt angle for various orientation distributions

To help guide users in defining an optimal tilt angle for data collection, we sought to provide a comprehensive and quantitative analysis of how different orientation distributions affect the SCF. We performed calculations according to two distinct scenarios, distinguished by the nature of the particle orientation distributions. The first scenario was characterized by a single conical distribution, wherein projections were constrained within a single cone defined by a half-angle of either 15°, 30°, 45°, or 60° degrees. This scenario simulates the case in which there is only one preferential orientation of the specimen. The second scenario was identical, except that we added a second set of projections, orthogonal to the first and characterized by an identical conical distribution. This scenario simulates the case in which there are two preferential orientations, related orthogonally to one another. To the orientation distribution for each of these scenarios, we then added randomly "sprinkled" projections outside of the primary conical orientation distribution. These sprinkled projections simulate the scenario in which a small subset of particles may reside within the ice layer, i.e., in between the two air-water interfaces, and are therefore free to adopt any orientation. For each scenario, we then simulated a series of data collections with a nominal stage tilt ranging from 10° to 60°, in 10° increments. We derived SCFs for all subsets, and we present the results in Fig. 10.

We first examined the most pathological scenario, wherein the projection distribution is narrowly confined to a cone of 15° (Fig. 10a). For such a distribution, the SCF is <0.81 in nearly all cases without remedial data-collection protocols. A 60° stage tilt, coupled with at least 20% sprinkled projections, is minimally required to yield a fully sampled reconstruction characterized by an SCF of at least 0.81. A lower stage tilt angle can be applied only if there are many projections randomly distributed outside of the primary projection cone. Next, if the orientations are distributed within a 30° cone, then applying a 60° stage tilt will lead to a fully sampled reconstruction, irrespective of the number of sprinkled projections. A smaller stage tilt can be applied, but the number of sprinkled projections outside of the main conical distribution must be at least 10% if a 50° stage tilt is used, and higher for smaller stage tilt angles. With a conical distribution of 45°, the application of a 50° stage tilt will now yield a fully sampled map. Furthermore, a smaller tilt angle can be used, such as 40° or even 30°, if a small amount of particle projections is sprinkled.

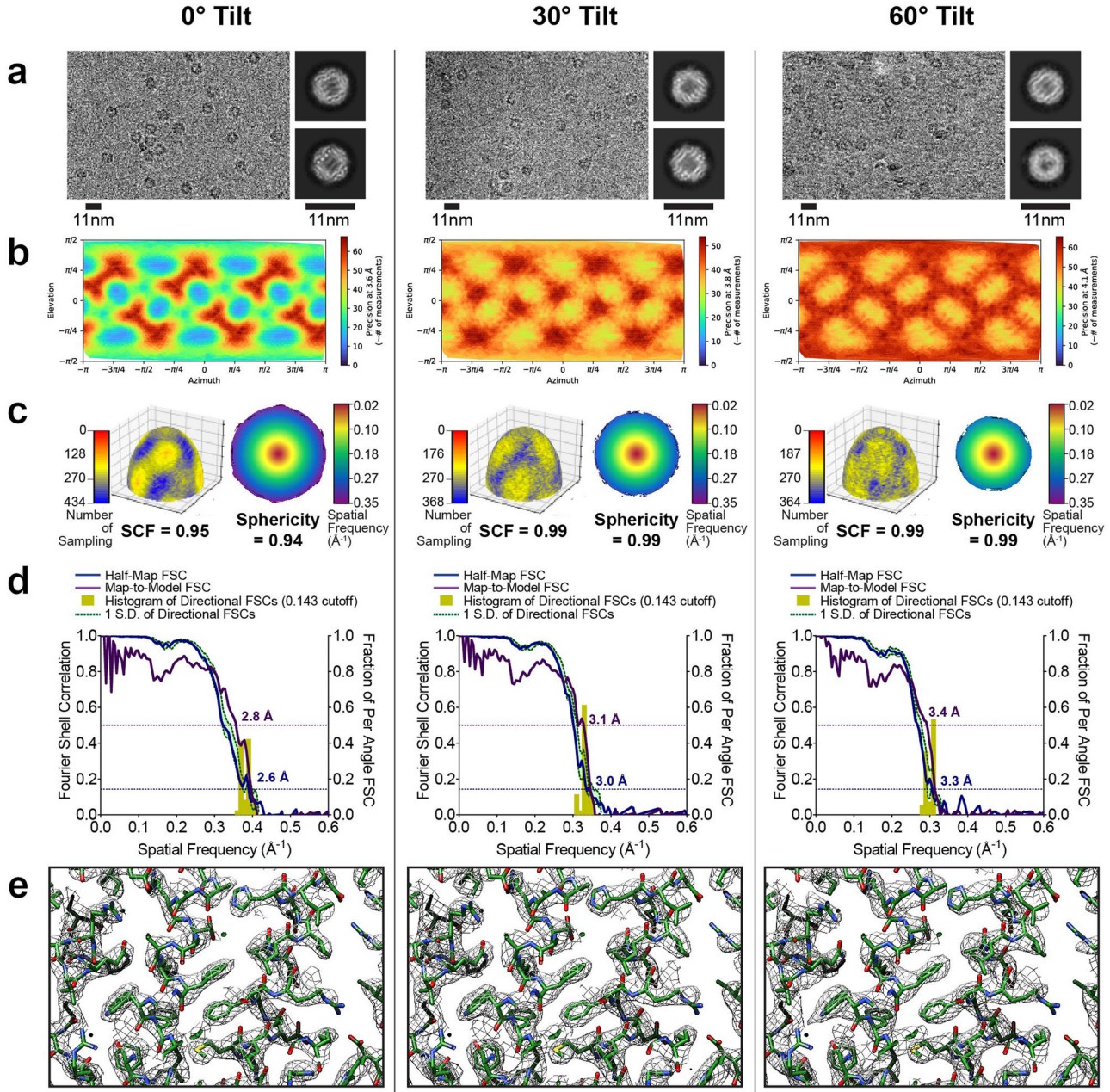

**Fig. 6 | Evaluation metrics for DPS at representative tilt angles.** Evaluation metrics of the cryo-EM data and reconstructed maps for the processing strategy outlined in Fig. 5. Results for untilted (left column), 30°-tilt (middle column) and 60°-tilt (right column) are compared. **a** Micrographs at 1.3 μm defocus, with corresponding 2D class averages at right. **b** Final Euler angle distribution. **c** Surface sampling plot with the sampling compensation factor (SCF) value below (left) and central slice through the 3DFSC colored by resolution (right). **d** Half-map and map-to-model global FSC curves. **e** Map density with docked atomic model.

Finally, if the projections are constrained to a 60° cone, the application of a 30° stage tilt will yield a fully sampled map, and smaller angles can be readily applied, depending on the number of sprinkled projections.

In the second scenario, representative of a sample possessing two orthogonally related conical distributions, the situation is considerably improved (Fig. 10b). Given a conical-distribution of 15°, a tilt angle of 40° will yield a fully sampled map, and in practice, a stage tilt angle of 30° can be used for most applications if even a small number of particles are oriented randomly outside of the two main distributions. With a 30° conical-distribution, the requisite tilt angle drops to 30°. Finally, for orthogonally populated sets with both 45°- and 60°- positioned distributions, tilting the stage is largely unnecessary, because both sets of distributions lead to fully (or nearly fully) sampled reconstructions, irrespective of the number of sprinkled projections.

## Discussion

In this work, using improved data processing methods, we extended the resolution of reconstructions from micrographs collected using stage tilt, to the point at which the maps derived from data collected with or without stage tilt are virtually indistinguishable. The improvement in global resolution is due to rigorous and iterative methods for refining CTF parameters and correcting for beam-induced motion. Although it is not practical to evaluate all available algorithms for data analysis, we envision that there are additional methodologies that can further extend the resolution of reconstructions from tilted images, and undoubtedly future advances will lead to even greater improvements.

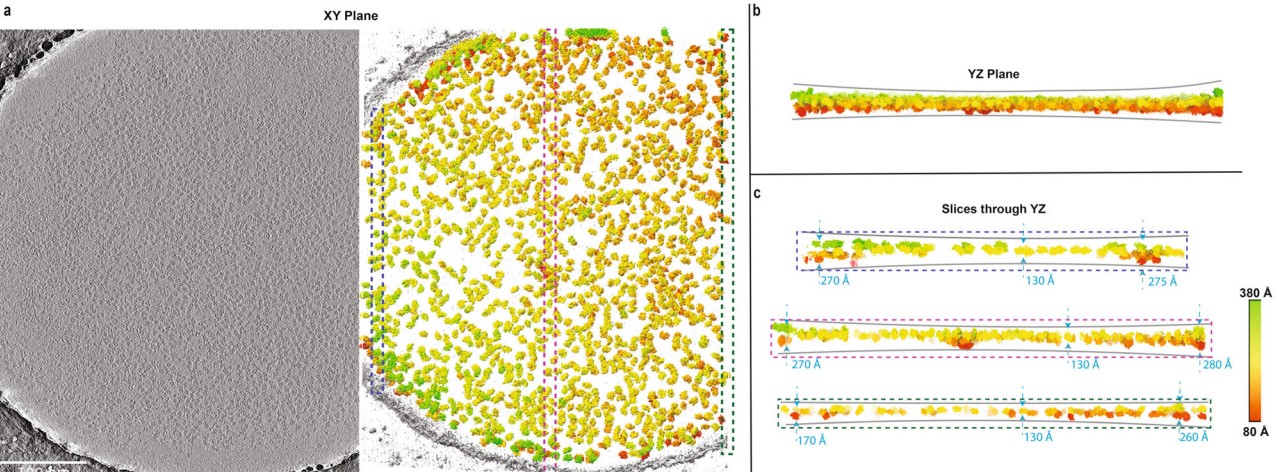

**Fig. 7 | Tomographic reconstruction of a tilt-series showing RNAP particles at the air-water interface. a** Representative slice through a tomogram showing the hole used for imaging (left) and rendering of particles within the same reconstructed tomogram viewed down the XY plane (right). **b** Cross-sectional view of tomogram along entirety of the YZ plane. **c** Thin slices of cross-sectional views at 3 distinct locations along the YZ-plane, with the top, middle and bottom slices representing three distinct locations of the hole. Dashed boxes in (**a**) display the locations of each slice in (**c**). Particles are colored according to their positions along the Z-axis, as extracted from the metadata file. Color key is described by the scale at the bottom-right panel.

When the stage is tilted, the amount of ice that the beam traverses become increasingly large as the tilt angle progresses towards 90°, because the effective ice thickness varies as 1/cos(tilt). At 30° stage tilt, the ice is thicker by ~15%, but at 60° stage tilt, the ice is thicker by 100% (i.e., twice). For higher tilt angles (~≥40°), the particle density in a given field of view may need to be optimized, because the effective projection area of a hole is reduced. This can be achieved by selecting areas containing more sparsely dispersed particles, such that when the stage is tilted, particles do not overlap. Thicker ice will lead to an increase in inelastic scattering that can only be partially corrected using energy filters[26]. Experimentally, the limits of ice thickness have recently been explored for single-particle cryo-EM reconstructions by Neselu et al. using apoferritin as a test system[41]. They showed that even with ice thickness ranging between 200-500 nm, it was possible to obtain a sub-3 Å reconstruction, although an energy filter and an acceleration voltage of 300 keV were both necessary. In the absence of an energy filter, the ice needed to be ~100–150 nm to yield sub-3 Å reconstructions. Independently, ice thickness limits for micro-electron diffraction have also been evaluated, with ice thickness of up to 360 nm showing no significant resolution loss for the resolution limit of around 2 Å[42]. Because most purified samples used for single-particle cryo-EM are characterized by ice that is less than ~100 nm in thickness[41], we suggest that ice thickness, per se, may not inherently limit the resolution. The more important consideration is whether the SNR for particles embedded within the ice layer is sufficient to assign correct orientations, as described below.

While ice thickness may not necessarily limit the attainable resolution, there is the potential problem of particle overlap. If there is significant particle overlap, then this might lead to greater inaccuracies in assigning orientation angles, yielding fewer good particles contributing to the high-resolution map. For example, we noted that more particle pruning was required for datasets collected using a large stage tilt angle (typically 40° or greater), especially for RNAP. This is, again, due to the geometry of the imaging experiment. Our general recommendation is to pre-screen grids prior to data collection using the expected tilt angle that will be used for data collection to ensure that particle overlap is minimized.

As the stage tilt angle is increased, beam induced motion becomes progressively larger, especially in the first few movie frames of data acquisition. Because the first ~2-5 e-/Å² carry the most high-resolution information, it is important to design the data collection and processing schemes so as to maximize the information from the experiment. It is possible to reduce beam-induced motion by using gold substrate grids[36] and to computationally correct for residual beam-induced motion recorded within the movies[35,43–45]. A combination of these two techniques may be necessary to overcome a large amount of cumulative motion, especially within the initial frames of the recorded movies. Although these steps have become routine even for standard untilted data collection, they are especially important for data collection schemes that employ stage tilting.

During computational data analysis, the defocus values for each particle are not known a priori and must be estimated from the data[18,46]. The closer the estimate is to the true defocus value, the better the correction, and hence the higher the resolution of the reconstruction[47]. In the absence of stage tilt, the defocus values for particles have a small spread. However, when the grid is tilted, there is a focus gradient in the field of view in addition to variations in focus due to the particle positions. Variations in focus in the image could be estimated using local CTF estimation approaches, but these typically rely on smaller regions of the image; the smaller fields of view may give rise to increased error in the estimates. There are numerous improved algorithms[32,48–52] that have led to improvements to the accuracy of local CTF estimation. In our experience working with different datasets, all the improved software implementations have yielded improvements to the final resolution estimates of both untilted and tilted datasets. As shown here, the improvements to resolution are particularly pronounced for tilted datasets (Figs. 1, 3, 5, and Supplementary Fig. 2 and 5). While we demonstrate one specific workflow in our current approach that employs the RELION and cryoSPARC softwares, there are likely alternative approaches that can yield similar improvements.

We did not attempt to quantify or evaluate the resolution attenuation associated with every possible parameter for data collection and/or image processing, because we view this as a moving target. As the methods improve and the field matures, the attenuation will become smaller. Ultimately, the experimentalist cares about the signal-to-noise ratio (SNR) of the final particle images, and how a drop in SNR can affect one's ability to refine a map reconstruction. The larger the particle, the higher the SNR, and the less resolution attenuation there will be as the stage is tilted. It seems that for a particle the size of AAV2, the decrease in SNR is not sufficient to impact the ability to align particles to significantly attenuate resolution for maps resolved to ~2 Å,

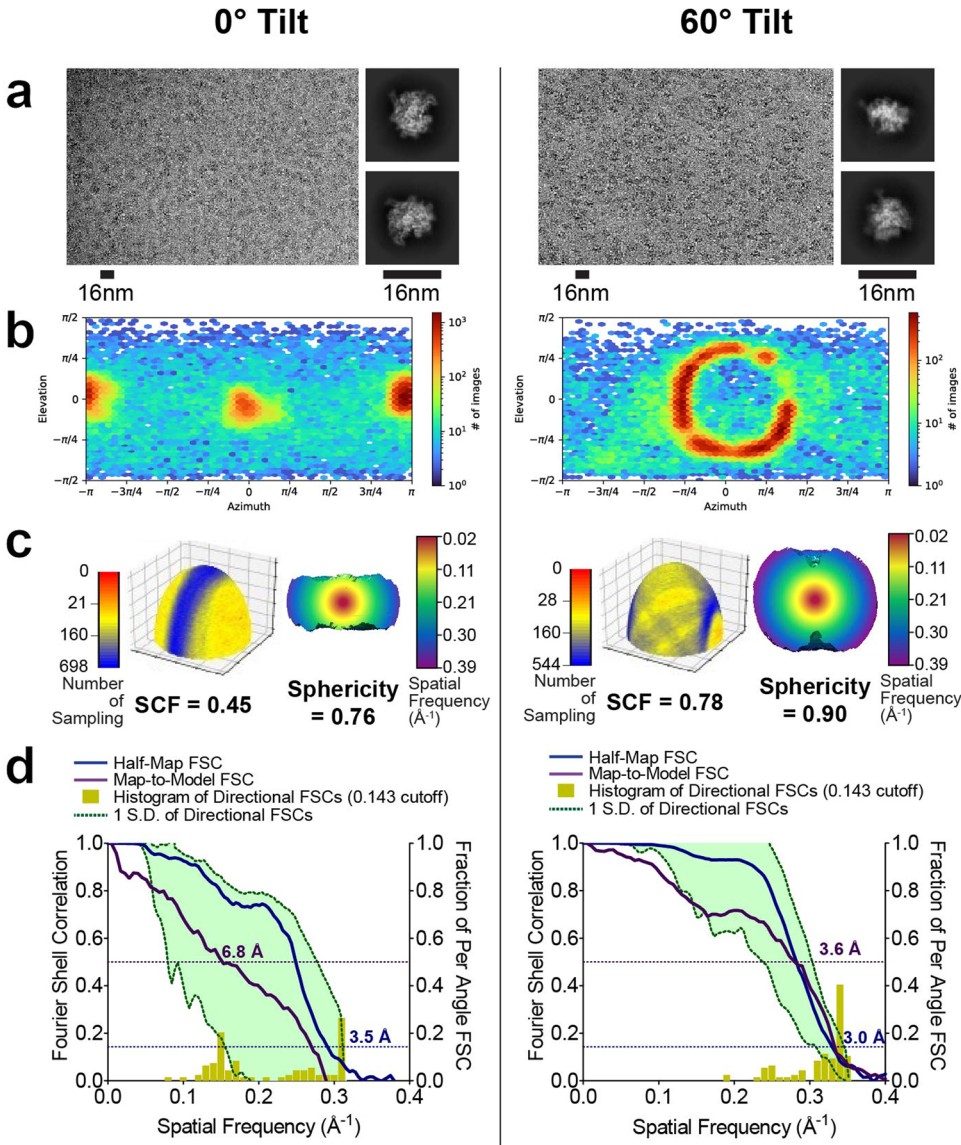

**Fig. 8 | Evaluation metrics for RNAP at 0° and 60° tilt angles.** RNAP data was collected without tilts and at 60° tilt angle. **a** Micrographs at 1.3 μm defocus, with corresponding 2D class averages at right. **b** Final Euler angle distributions. **c** Surface sampling plots with the sampling compensation factor (SCF) value below (left) and central slice through 3DFSC, colored by resolution. **d** Half-map and map-to-model global FSC curves.

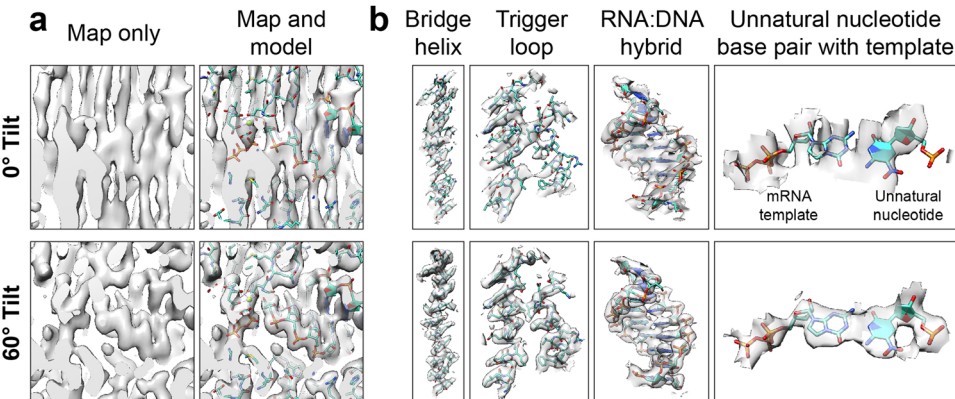

**Fig. 9 | Map and model of RNAP showing unnatural base pair recognition.** **a** RNAP maps from data collected without tilt (top) and with 60°-tilt (bottom) are shown, with (right) and without (left) the superimposed atomic model. The map features from the reconstruction from 0° data are elongated and cannot be properly interpreted with an atomic model. In contrast, the map features from the reconstruction from 60° data are consistent with the expected densities for the chemical groups. **b** Map features that of RNAP that can be clearly observed within the reconstructions from 60° tilted data include the bridge helix, the trigger loop, the RNA:DNA hybrid, and an unnatural base-pair within the enzymatic active site.

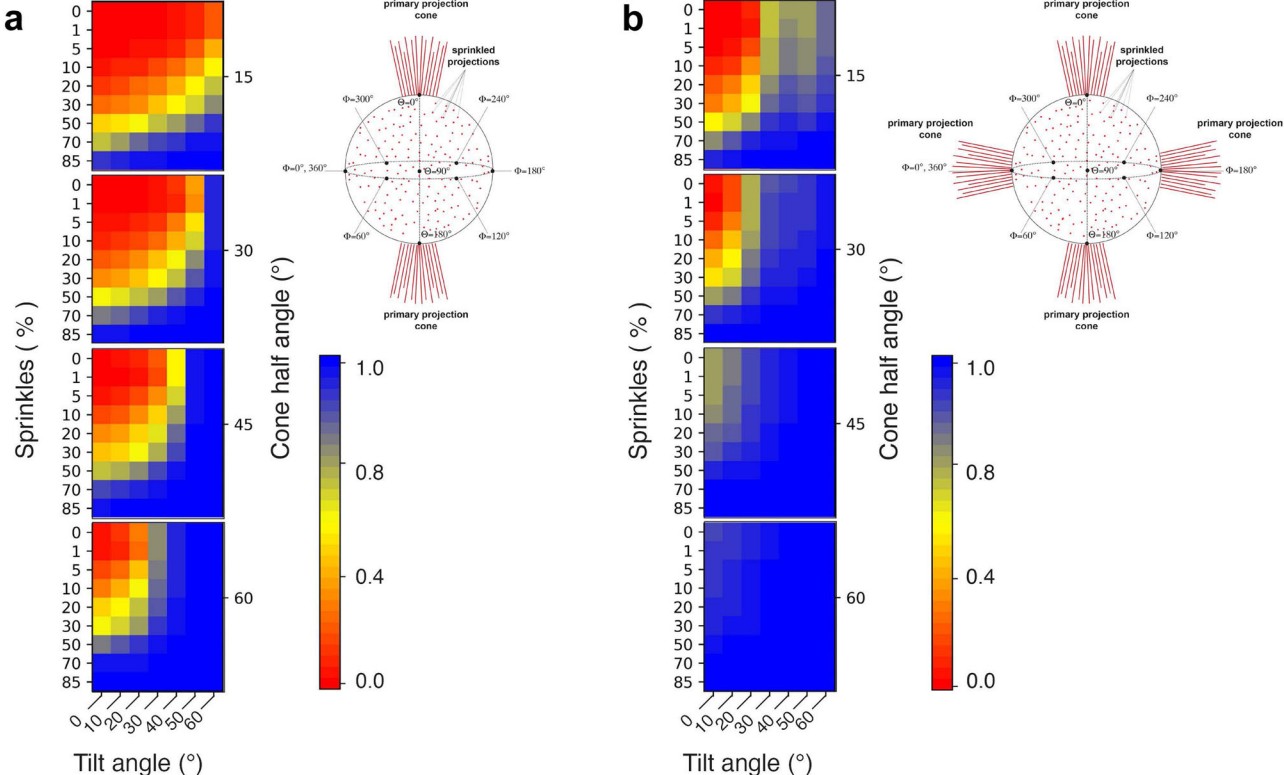

**Fig. 10 | Quantitative assessment of how tilting improves Fourier sampling.** The value of the SCF is calculated for datasets (**a**) either with a single preferred orientation (schematic on top right) or (**b**) two preferred orientations orthogonal to another (schematic on top right). The sampling compensation factor (SCF) for a given tilt angle is calculated as a function of both the percentage of other views (sprinkles) and the spread of the preferred orientation (cone half angle). The SCF value range is displayed from 0.0 (red) to 1.0 (blue) (bottom right in both **a** and **b**).

even at tilt angles as high as 60° (Figs. 1–2). Of note, our AAV2 data was collected using the previous generation K2 Summit direct electron detector, whereas newer detectors, such as the K3 or Falcon4, have improved detective quantum efficiencies and allow the experimentalist to use higher frame rates to correct for beam-induced movement. For particles the size of apoferritin, the data processing strategies that we have employed are, in most cases (except for the 60° tilt angle) sufficient to obtain global resolutions for tilted datasets that are similar to or nearly indistinguishable from their respective untilted counterparts (Figs. 3–4). Moreover, we show that sub-2 Å resolution reconstructions are attainable from 30°-tilted images, and this reconstruction is virtually indistinguishable from a reconstruction from untilted images (Supplementary Fig. 1). For smaller proteins, tilting the stage apparently leads to more information loss, because there is a drop in relative SNR of the particles. Current methods may not fully recover high-resolution features in comparison to reconstructions from untilted data collections (Figs. 5–6). Nevertheless, stage tilt has been employed for proteins as small at ~43 kDa to improve map quality[53], including membrane proteins in the ~100 kDa range[54,55]. Possible methods to improve the SNR of images containing small proteins include: the use of denoising algorithms to boost the low to intermediate resolution spatial frequency ranges[56,57], and thus facilitate orientation alignment; the use of electron detectors with better DQE; the use of novel grid substrates, such as HexAufoil[58]. In the case of HexAufoil grids, even when the stage is tilted, the apparent root mean square displacement of specimen motion is comparable to untilted data collection[58]. These strategies, along with the data processing pipeline described here, may be required for tilted data collection of smaller proteins suffering from pathological preferred orientation.

We applied our methods to the structure determination of RNAP, which contains a bound unnatural base (Figs. 8–9). In the

reconstruction, we could clearly resolve important features of the RNAP elongation complex, such as base stacking of upstream RNA:DNA hybrid, and the side chains of the bridge helix. Intriguingly, we also observe clear undistorted density corresponding to the incoming nucleotide substrate, which binds at the canonical addition site and density attributed to the trigger loop motif[59]. The model derived from the map revealed that unnatural substrate can bind to the same nucleotide addition site as natural substrate.

Finally, we demonstrate a practical framework for quantifying the amount of stage tilt that would be required to overcome resolution anisotropy inherent to preferentially oriented specimens (Fig. 10). This graphical representation visually demonstrates how the SCF changes when there are additional views or when stage tilt is applied, and complements the previously described software to explicitly measure the SCF of tilted data given any orientation distribution[14]. The SCF values derived from varying (i) the cone size of the projection distribution, (ii) the number of sprinkled projections outside the cone, or (iii) the tilt angle applied to the dataset, are consistent with experiment, as evidenced using the dataset of RNAP. In the absence of stage tilt, the orientations for RNAP are distributed within a cone of ~20–25°, with a small number of sprinkled projections. We found that collecting data with the stage tilt angle set to 60° led to a nearly, but still incompletely sampled reconstruction, characterized by an SCF of 0.78. The fact that the SCF was just shy of the suggested baseline of 0.81 means that the primary projection distribution is <30° and there are few true positive orientations outside of the primary projection distribution. Our initial predicted optimal stage tilt angle was 51°, which should have led to an orientation distribution characterized by an SCF of 0.81. In practice, we found that a slightly higher stage tilt angle was necessary to yield a (nearly) fully sampled map. Importantly, the features of the map from data collected at 60° stage tilt were of high-quality, allowing us

to derive an accurate atomic model and highlight distinct aspects of RNAP biology. We attribute the overly optimistic initial SCF prediction to the inclusion of extraneous false-positive angular orientation assignments. Currently, there are no effective methods to quantify or eliminate false-positive angular assignments in cryo-EM data, and hence the user is advised to apply a slightly higher tilt angle than what is predicted. Collectively, these results provide a quantitative, graphical representation for the SCF of cryo-EM reconstructions. Our SCF analyses will help users define an optimal tilt angle for any dataset characterized by preferred orientation.

## Methods

### Statistics
For calculations of Fourier shell correlations (FSC), the FSC cut-off criterion of 0.143[60] was used.

### AAV2 production, purification and vitrification
AAV2 were produced utilizing the Bac-to-bac system (Thermo Fisher) as previously described[31]. Briefly, recombinant baculoviruses encoding the AAV2 cap gene were used to infect Sf9 insect cells (ATCC, Cat#CRL-1711) at a multiplicity of infectivity of 5. The infected cells were harvested 72 hours post infection and lysed by three freeze-thaw cycles. Following Benzonase (EMD Millipore Cat#712053) treatment the lysates were clarified and purified using a step iodixanol gradient with subsequent anion exchange chromatography. The sample was dialyzed into 50 mM HEPES, pH 7.4 with 2 mM $MgCl_2$ and 150 mM NaCl and concentrated with an Apollo 7-ml centrifugal concentrator (Orbital Biosciences, Topsfield, MA).

To increase particle concentration, double blotting was done[61]. UltrAuFoil R 1.2/1.3 Au 300 mesh gold grids (Quantifoil) were used for cryo-EM sample preparation. The grids were plasma treated with the Solarus Plasma Cleaner 950 (Gatan, 75% argon/25% oxygen atmosphere at 15 Watts for 7 s) immediately before sample application to make their surfaces hydrophilic. A 2.5 µL aliquot of AAV2 sample at 2.5 mg mL$^{-1}$ concentration was applied to grids and blotted away using filter paper after 20 s wait time. Another 2.5 µL of the same sample was then re-applied to the grid and blotted after 20 s wait time and then vitrified in liquid ethane using a manual plunger. All operations were performed in a 4 °C cold room at >80% humidity to minimize evaporation and sample degradation.

### AAV2 cryo-EM data collection and processing
Images were recorded on a Titan Krios electron microscope (FEI/Thermo Fisher) equipped with a K2 Summit direct detector (Gatan) at 1.026 Å per pixel in counting mode using the Leginon software package[62–66]. Data collection was performed using a dose of ~78.0 e⁻ Å$^{-2}$ across 200 frames (50 msec frame$^{-1}$) at a dose rate of ~8.2 e⁻ pix$^{-1}$ s, using a set nominal defocus range of −0.8 to −2 µm with 100 µm objective aperture inserted. A total of 1229 micrographs was recorded over a single 4-day data collection using stage shift, across tilt angles of 0°, 10°, 20°, 30°, 40°, 50° and 60° set with stage alpha. We note that the AAV2 data collection, and all other subsequent data collections, were not performed continuously, because we were concerned with having identical conditions for accurate comparisons between different stage tilt angles. Modern workflows for data collection can yield higher throughput, especially using beam-tilt induced image-shift and active beam-tilt compensation for targeting[67]. Once the tilt angle is defined and the imaging conditions are established, the throughput for collecting data using stage tilt should rival that of collecting data without stage tilt.

Movie frames were aligned using MotionCor2[43] with 7 by 7 patches, a grouping of 3 and B-factor of 500 for global alignment and 150 for the patches, initially through Appion[68], then through RELION[34,51]. Micrograph CTF estimation was performed using both CTFFind4[46] for whole micrographs and GCTF[32] for individual particles within the

Appion software package. A subset of eight micrographs was first used for particle picking using Gautomatch (Kai Zhang, unpublished, https://www2.mrc-lmb.cam.ac.uk/download/gautomatch-053/), and particles were extracted and analyzed by 2D classification in RELION. 2D class averages that showed clear structural details were used as templates for template-based picking using Gautomatch on all 1229 micrographs. All particles were then extracted using a box size of 600 pixels and subjected to two initial rounds of 2D classification (binned by 4) to identify and discard false positives such as ice and other obvious contaminants. Following 2D classification, 66,763 particles were re-extracted with the re-centering option in RELION. Finally, the particles from all datasets were again equalized to ensure the distributions of defocus values were consistent across tilts (Supplementary Table 1), resulting in 7000 particles per tilt angle.

With the equalized dataset of 7000 particles per tilt angle, cryoSPARC homogeneous refinement was first performed using cryoSPARC ab initio map generated from the dataset. Thereafter, cryoSPARC Patch CTF[50] was performed, followed by RELION CTF refinement[52], RELION Bayesian polishing[52] (using v/d/a of 0.5/12000/1), second round of RELION CTF refinement, second round of RELION Bayesian polishing (using v/d/a of 0.5/12000/1), third round of RELION CTF refinement and finally cryoSPARC Ewald sphere curvature correction[50]. In between these refinements, a cryoSPARC homogeneous refinement was performed in order to obtain the refined map, Euler angles and shifts that would be used for the next refinement. All conversions between RELION and cryoSPARC were performed using Daniel Asarnow's pyem script[69]. While developing this workflow with AAV2, we compared the RELION and cryoSPARC implementations of CTF refinement. We did not observe any difference in resolution improvement when CTF refinement was performed in RELION versus cryoSPARC for the AAV2 dataset. Thus, for subsequent datasets, we used the cryoSPARC implementation, because it was simpler to implement immediately after refinement and did not require additional conversion steps.

For the map-to-model analysis, the PDB-6E9D of the same AAV2 variant was used. The PDB was converted to a map using molmap function in UCSF Chimera[70]. The mask was generated using this PDB initial model which was binarized at 0.01 threshold, extended by 3 pixels, given a soft Gaussian edge of 3 pixels using 'relion_mask_create'. The sharpened AAV2 maps were then compared against the aligned PDB generated maps, using the PDB generated mask, to obtain the map-to-model FSC. EMAN[71] was used to generate these FSC curves. SCF values were generated using the SCF command line tool[13,14], while 3DFSC results were generated using the 3DFSC web server[15]. Molecular graphics were generated using UCSF Chimera[70].

X and Y coordinates of the equalized particles across tilt angles were plotted on a histogram to confirm that there was no preferential distribution of the particles on the tilt axis at high tilt angles (Supplementary Fig. 8).

### Apoferritin purification and vitrification
For data collected at Scripps Research, apoferritin was purchased (Thermo Fisher Scientific) at a concentration of approximately 8 µM (3.5–4.0 mg mL$^{-1}$). Two dilutions were prepared having final concentrations of ~1.6 and 0.8 µM in a buffer containing 50 mM Tris-HCl pH 7.5, 300 mM NaCl 0.5 mM TCEP. Untilted, 10° and 20° tilted datasets were collected with grids prepared from ~1.6 µM apoferritin and the rest of the datasets were collected with grids prepared from ~0.8 µM apoferritin.

UltrAuFoil R 1.2/1.3 Au 300 mesh gold grids (Quantifoil) were used for cryo-EM sample preparation. The grids were plasma treated with the Solarus Plasma Cleaner 950 (Gatan, 75% argon/ 25% oxygen atmosphere at 15 Watts for 7 s) immediately before sample application to make their surfaces hydrophilic. A 3 µL aliquot of apoferritin sample

at ~1.6 or 0.8 μM concentration was applied to grids and then vitrified in liquid ethane using a manual plunger. All operations were performed in a 4 °C cold room at >80% humidity to minimize evaporation and sample degradation.

For data collected at Janelia Research Campus, apoferritin was purified as described before[72]. Grids were vitrified in a manner similar to the grids prepared for data collection at Scripps Research.

## Apoferritin cryo-EM data collection (Scripps) and processing

Images were recorded on a Titan Krios electron microscope (FEI/ Thermo Fisher) equipped with a K3 BioQuantum direct detector (Gatan) at 0.830 Å per pixel in counting mode (CDS enabled) using the Leginon software package[62–66]. Energy filter slit width of 20 eV was used. Data collection was performed using a dose of ~44.2 e$^-$ Å$^{-2}$ across 100 frames (15 msec frame$^{-1}$) at a dose rate of ~20.3 e$^-$ pix$^{-1}$ s, using a set nominal defocus range of −0.8 to −2 μm with 100 μm objective aperture inserted. A total of 637 micrographs was recorded over a single 4-day data collection using stage shift, across tilt angles of 0°, 10°, 20°, 30°, 40°, 50° and 60° set with stage alpha.

Movie frames were aligned using MotionCor2[43] with 7 by 5 patches, a grouping of 2 and B-factor of 500 for global alignment and 150 for the patches, through RELION[34,51]. Micrograph CTF estimation was performed using both CTFFind4[46] for whole micrographs and GCTF[32] for individual particles within the Appion software package. Template picking using previous apoferritin 2D class averages from routine apoferritin test runs was done. All particles were then extracted using a box size of 320 pixels and subjected to two initial rounds of 2D classification (binned by 4, then by 2) to identify and discard false positives such as ice and other obvious contaminants. Finally, the particles from all datasets were again equalized to ensure the distributions of defocus values were consistent across tilts (Supplementary Table 2), resulting in 17,000 particles per tilt angle.

With the equalized dataset of 17,000 particles per tilt angle, cryoSPARC homogeneous refinement was first performed using cryoSPARC ab initio map generated from the dataset. Thereafter, cryoSPARC Patch CTF[50] was performed, followed by cryoSPARC CTF refinement[50], RELION Bayesian polishing[52] (using v/d/a of 0.5/12000/1), second round of cryoSPARC CTF refinement, second round of RELION Bayesian polishing (using v/d/a of 0.5/12000/1), third round of cryoSPARC CTF refinement and finally cryoSPARC higher order aberration correction[50]. In between these refinements, a cryoSPARC homogeneous refinement was performed in order to obtain the refined map, Euler angles and shifts that would be used for the next refinement. All conversions between RELION and cryoSPARC were performed using Daniel Asarnow's pyem script[69].

For the map-to-model analysis, the PDB-7A6A of apoferritin was used. The PDB was converted to a map using molmap function in UCSF Chimera[70]. The mask was generated using this PDB initial model which was binarized at 0.01 threshold, extended by 3 pixels, given a soft Gaussian edge of 3 pixels using 'relion_mask_create'. The sharpened apoferritin maps were then compared against the aligned PDB generated maps, using the PDB generated mask, to obtain the map-to-model FSC. RELION was used to generate these FSC curves. SCF values were generated using the SCF command line tool[13,14], while 3DFSC results were generated using the 3DFSC web server[15]. Molecular graphics were generated using UCSF Chimera[70].

X and Y coordinates of the equalized particles across tilt angles were plotted on a histogram to confirm that there was no preferential distribution of the particles on the tilt axis at high tilt angles (Supplementary Fig. 9).

## Apoferritin cryo-EM data collection (Janelia) and processing

Images were recorded on a Titan Krios electron microscope (FEI/ Thermo Fisher) equipped with a K3 BioQuantum direct detector (Gatan) at 0.830 Å per pixel in counting mode (CDS mode) using

SerialEM[73]. Energy filter slit width of 20 eV was used. Data collection was performed using a dose of ~41.2 e$^-$ Å$^{-2}$ across 270 frames (15 msec frame$^{-1}$) at a dose rate of ~7 e$^-$ pix$^{-1}$ s, using an estimated defocus range of −0.2 to −1.7 μm for untilted and −0.1 to −2.8 μm for 30° tilted data collection with 100 μm objective aperture inserted. A total of 632 micrographs was recorded over a single 4-day data collection using stage shift, across tilt angles of 0° and 30° set with stage alpha.

Movie frames were aligned using MotionCor2[43] with 4 by 4 patches, a grouping of 1 and B-factor of 150 for both global alignment and patches through RELION[34,51]. Micrograph CTF estimation was performed using both cryoSPARC Patch CTF[50] for whole micrographs within the cryoSPARC GUI and GCTF[32] for individual particles using a command-line script. Template picking using previous apoferritin 2D class averages from routine apoferritin test runs was done. All particles were then extracted using a box size of 256 pixels and subjected to several rounds of 2D classification to identify and discard false positives such as ice and other obvious contaminants. Finally, the particles from all datasets were again equalized to ensure the distributions of defocus values were consistent across tilts (Supplementary Table 3), resulting in 64,000 particles per tilt angle.

With the equalized dataset of 64,000 particles per tilt angle, cryoSPARC homogeneous refinement was first performed using cryoSPARC ab initio map generated from the dataset. Thereafter, cryoSPARC CTF refinement[50] followed by RELION Bayesian polishing[52] (using v/d/a of 0.5/12000/1) were performed iteratively for three rounds. In between these refinements, a cryoSPARC homogeneous refinement was performed in order to obtain the refined map, Euler angles and shifts that would be used for the next refinement. All conversions between RELION and cryoSPARC were performed using Daniel Asarnow's pyem script[69].

For the map-to-model analysis, the PDB-7A6A of apoferritin was used. The PDB was converted to a map using molmap function in UCSF Chimera[70]. The sharpened apoferritin maps were then compared against the aligned PDB generated maps, using the mask generated during final cryoSPARC homogeneous refinement, to obtain the map-to-model FSC. SCF values were generated using the SCF command line tool[13,14], while 3DFSC and 2D FSC results were generated using the 3DFSC web server[15]. Molecular graphics were generated using UCSF Chimera[70].

## Proteasome vitrification

Proteasome was obtained as a gift from Yifan Cheng. UltrAuFoil R 1.2/ 1.3 Au 300 mesh gold grids (Quantifoil) were used for cryo-EM sample preparation. The grids were plasma treated with the Solarus Plasma Cleaner 950 (Gatan, 75% argon/25% oxygen atmosphere at 15 Watts for 30 s) immediately before sample application to make their surfaces hydrophilic. 3 uL of proteasome at 0.3 mg mL$^{-1}$ concentration was applied to grids and plunge-frozen using a Leica EM-grid plunger (Leica Microsystems), with the chamber humidity maintained at 90% humidity and 4 °C.

## Proteasome cryo-EM data collection and processing

Images were recorded on a Titan Krios electron microscope (FEI/ Thermo Fisher) equipped with a K2 Summit direct detector (Gatan) at 1.07 Å per pixel in counting mode using the Leginon software package[62–66]. Data collection was performed using a dose of ~93.8 e$^-$ Å$^{-2}$ across 70 frames (200 msec frame$^{-1}$) at a dose rate of ~7.7 e$^-$ pix$^{-1}$ s, using a set nominal defocus range of −1.8 to −2.8 μm with 100 μm objective aperture inserted. A total of 420 micrographs was recorded over a single 2-day data collection using stage shift, across tilt angles of 0°, 10°, 20°, 30°, 40° and 50° set with stage alpha.

Movie frames were aligned using MotionCor2[43] with 7 by 7 patches, a grouping of 1 and B-factor of 150 for global alignment and 100 for the patches, through RELION[34,51]. Micrograph CTF estimation was performed using both CTFFind4[46] for whole micrographs and GCTF[32]

for individual particles within the Appion software package. Template picking using previous apoferritin 2D class averages from routine proteasome test runs was done. All particles were then extracted using a box size of 400 pixels and subjected to two initial rounds of 2D classification (binned by 4, then by 2) to identify and discard false positives such as ice and other obvious contaminants. Finally, the particles from all datasets were again equalized to ensure the distributions of defocus values were consistent across tilts (Supplementary Table 4), resulting in 5000 particles per tilt angle.

With the equalized dataset of 5000 particles per tilt angle, cryoSPARC homogeneous refinement was first performed using cryoSPARC ab initio map generated from the dataset. Thereafter, cryoSPARC Patch CTF[50] was performed, followed by cryoSPARC CTF refinement[50], RELION Bayesian polishing[52] (using v/d/a of 0.5/12000/1) and final round of cryoSPARC CTF refinement. In between these refinements, a cryoSPARC homogeneous refinement was performed to obtain the refined map, Euler angles and shifts that would be used for the next refinement. All conversions between RELION and cryoSPARC were performed using Daniel Asarnow's pyem script[69].

For the map-to-model analysis, the PDB-1YAR of proteasome was used. The PDB was converted to a map using molmap function in UCSF Chimera[70]. The mask was generated using this PDB initial model which was binarized at 0.01 threshold, extended by 3 pixels, given a soft Gaussian edge of 3 pixels using 'relion_mask_create'. The sharpened proteasome maps were then compared against the aligned PDB generated maps, using the PDB generated mask, to obtain the map-to-model FSC. RELION was used to generate these FSC curves. SCF values were generated using the SCF command line tool[13,14], while 3DFSC results were generated using the 3DFSC web server[15]. Molecular graphics were generated using UCSF Chimera[70].

## DPS purification and vitrification

A modified pET15b expression plasmid for *E. coli* DPS, which yields the target protein fused to an N-terminal hexahistidine tag and TEV-protease cleavage site, was provided by Christopher Russo (MRC-LMB). DPS protein was heterologously over-expressed in the *E. coli* BL21(DE3)-RIPL (Agilent) expression host. *E. coli* cultures were grown at 37 °C in TB medium (supplemented with 50 µg/ml ampicillin) to an optical density (600 nm) of 1.5, induced with 0.5 mM isopropyl-β-D-thiogalactoside, and allowed to grow for an additional 16 h at 18 °C. Bacterial cells were harvested by centrifugation, resuspended in lysis buffer (50 mM Tris-HCl, pH 8.0; 0.5 M NaCl; 20 mM imidazole; 1% v/v Tween20; 10% v/v glycerol; and 10 mM 2-mercaptoethanol), and lysed by sonication. The DPS protein was isolated from the lysate by affinity chromatography with nickel nitrilotriacetic-acid-coupled agarose (Thermo Fisher Scientific) and eluted with lysis buffer containing 0.25 M imidazole. The partially purified DPS protein (His₆-tag uncleaved) was then further purified by gel-filtration chromatography on a Superdex 200 HR26/60 column (Amersham Biosciences), with a column buffer containing 50 mM Tris-HCl, pH 8.0, 0.5 M NaCl, and 10 mM 2-mercaptoethanol. The final purified DPS protein was dialyzed into a buffer containing 12.5 mM Tris-HCl, pH 8.0, 0.05 M NaCl, and 10 mM 2-mercaptoethanol, and then concentrated to ~10 mg/ml. For estimation of the size of the DPS oligomer by size-exclusion chromatography, the elution volume of DPS was compared to that of a panel of other proteins with a similar or smaller oligomer size.

UltrAuFoil R 1.2/1.3 Au 300 mesh gold grids (Quantifoil) were used for cryo-EM sample preparation. The grids were plasma treated with the Solarus Plasma Cleaner 950 (Gatan, 75% argon/ 25% oxygen atmosphere at 15 Watts for 7 s) immediately before sample application to make their surfaces hydrophilic. A 2.5 µL aliquot of DPS sample at 0.0625 mg mL⁻¹ concentration was applied to grids and then vitrified in liquid ethane using a manual plunger. All operations were performed

in a 4 °C cold room at >80% humidity to minimize evaporation and sample degradation.

## DPS cryo-EM data collection and processing

Images were recorded on a Titan Krios electron microscope (FEI/Thermo Fisher) equipped with a K3 BioQuantum direct detector (Gatan) at 0.834 Å per pixel in counting mode (CDS enabled) using the Leginon software package[62–66]. Energy filter slit width of 20 eV was used. Data collection was performed using a dose of ~59.6 e⁻ Å⁻² across 195 frames (13 msec frame⁻¹) at a dose rate of ~16.6 e⁻ pix⁻¹ s, using a set nominal defocus range of −1.2 to −2.0 µm with 100 µm objective aperture inserted. A total of 194 micrographs were recorded over a single 2-day data collection using stage shift, across tilt angles of 0°, 10°, 20°, 30°, 40°, 50° and 60° set with stage alpha.

Movie frames were aligned using MotionCor2[43] with 7 by 5 patches, a grouping of 1 and B-factor of 500 for global alignment and 150 for the patches, through RELION[34,51]. Micrograph CTF estimation was performed using both CTFFind4[46] for whole micrographs and GCTF[32] for individual particles within the Appion software package. CryoSPARC blob picker was used to pick particles from all micrographs, and an initial 2D classification was done to generate templates for template picking. All template picked particles were then extracted using a box size of 256 pixels and subjected to two initial rounds of 2D classification (binned by 4, then by 2) to identify and discard false positives such as ice and other obvious contaminants. Finally, the particles from all datasets were again equalized to ensure the distributions of defocus values were consistent across tilts (Supplementary Table 5), resulting in 1750 particles per tilt angle.

With the equalized dataset of 1750 particles per tilt angle, cryoSPARC homogeneous refinement was first performed using cryoSPARC ab initio map generated from the dataset. Thereafter, cryoSPARC Patch CTF[50] was performed, followed by RELION Bayesian polishing[52] (using v/d/a of 1/7000/0.5), second round of RELION Bayesian polishing[52] (using v/d/a of 1/7000/0.5), cryoSPARC CTF refinement[50], and finally higher order aberration correction using RELION[52]. In between these refinements, a cryoSPARC homogeneous refinement was performed to obtain the refined map, Euler angles and shifts that would be used for the next refinement. All conversions between RELION and cryoSPARC were performed using Daniel Asarnow's pyem script[69].

For the map-to-model analysis, the PDB- 6GCM of DPS was used. The PDB was converted to a map using molmap function in UCSF Chimera[70]. The mask was generated using this PDB initial model which was binarized at 0.01 threshold, extended by 3 pixels, given a soft Gaussian edge of 3 pixels using 'relion_mask_create'. The sharpened proteasome maps were then compared against the aligned PDB generated maps, using the PDB generated mask, to obtain the map-to-model FSC. RELION was used to generate these FSC curves. SCF values were generated using the SCF command line tool[13,14], while 3DFSC results were generated using the 3DFSC web server[15]. Molecular graphics were generated using UCSF Chimera[70].

X and Y coordinates of the equalized particles across tilt angles were plotted on a histogram to confirm that there was no preferential distribution of the particles on the tilt axis at high tilt angles (Supplementary Fig. 10).

## RNAP purification and vitrification

The expression plasmids for *E. coli* RNA polymerase (deposited by Seth Darst lab) were purchased from Addgene. We followed previous purification methods[74] with modification. pEcRNAP6 (Addgene # 128940) and pACYCDuet-LEc-rpoZ (Addgene # 128837) were co-transformed to BL21 (DE3) competent cells. 0.25 mM of IPTG was added to cell culture when O.D. reached 0.6 and protein were expressed in 20 °C for overnight. Cells were collected and resuspended in Buffer A [50 mM Tris (pH 7.4), 500 mM NaCl, 5% (v/v) glycerol, 2 mM β-mercaptoethanol

(BME)] with 1X EDTA-free protease inhibitor (Sigma-Aldrich). After lysis and centrifugation, cell lysate was loaded to Ni-NTA resin and washed with Buffer A + 20 mM imidazole, Buffer A (300 mM NaCl) + 30 mM imidazole and eluted by Buffer A (300 mM NaCl) + 250 mM imidazole. Eluted sample was diluted with same volume of Buffer A (0 mM NaCl) to reduce NaCl concentration to 150 mM. After dilution, sample was loaded onto HiTrap Heparin column, and purified by increasing NaCl concentration from 150 mM to 800 mM. For anion exchange purification, heparin elute was dialyzed against Buffer B (20 mM HEPES pH 8.0, 100 mM NaCl, 5% (v/v) glycerol, 1 mM DTT). After dialysis, the sample was loaded into HiTrap Q column. *E. coli* RNA polymerase ($\alpha_2\beta'\beta\omega$) was purified by increasing NaCl concentration from 100 mM to 800 mM. During concentration, buffer was changed to Buffer B with 150 mM NaCl, flash frozen in liquid nitrogen and stored in −80 °C for future use.

For vitrification, mini-scaffold containing RNA 5′- AUCGAGAGG −3′, tsDNA 5′- CCTTCTCTCTCTCGCTGA(dZ)CCTCTCGATG −3′ where dZ is 6-amino-3-(1′-β-D-2′-deoxyribofuranosyl)−5-nitro-1H-pyridin-2-one (AEGIS Phosphoramidite dZ from Firebird Biomolecular Sciences LLC, Alcachua, FL, catalog No. dZ-PA-101) and ntsDNA (5′- TCAGCGAGAGA GAGAAGG-3′) with molar ratio of 1.2:1:1.2 were annealed in 1X elongation buffer (EB, 20 mM Tris pH 7.5, 40 mM KCl, 5 mM MgCl₂, 5 mM DTT). To form the elongation complex, purified *E. coli* RNAP were mixed with prepared Mini-scaffold with molar ratio 1:1.3 and incubated in ice for 1 hour. Final 4 mM of MgCl₂ and rPTP where rP is 2-amino-8-(1′-β-D-ribofuranosyl)-imidazo-[1,2a]−1,3,5-triazin-[8H]−4-one (not 2′-deoxy) (AEGIS ribotriphosphate rPTP from Firebird Biomolecular Sciences LLC, Alachua, FL) in 1X EB were added to elongation complex and incubated in ice for 30 min. The final *E. coli* RNAP concentration was 15 - 18 mg/ml. If dilution was needed, 4 mM substrate nucleotide triphosphate and 4 mM MgCl₂ in 1X EB were used.

UltrAuFoil R 1.2/1.3 Au 300 mesh gold grids (Quantifoil) were used for cryo-EM sample preparation. The grids were plasma treated with the Solarus Plasma Cleaner 950 (Gatan, 75% argon/25% oxygen atmosphere at 15 Watts for 7 s) immediately before sample application to make their surfaces hydrophilic. Grids were prepared using a home-made manual plunging apparatus in a cold room where the temperature was between 3–5 °C and humidity was in excess of 85%. A range of 0.05−0.4 mg ml⁻¹ of sample concentration was used to prepare grids of different concentration.

## RNAP cryo-EM data collection and processing

Images were recorded on a Titan Krios electron microscope (FEI/Thermo Fisher) equipped with a K3 BioQuantum direct detector (Gatan) at 0.83 Å per pixel in counting mode (CDS enabled) using the Leginon software package[62–65]. Energy filter slit width of 20 eV was used. Movies were recorded using a beam-image shift targeting strategy with the 100 μm objective aperture inserted.

In order to initially gauge the number of micrographs/particles that would be needed to exceed ~4 Å resolution, on-the-fly data processing using cryoSPARC Live was performed. For untilted specimen stage tilt, a concentration of 0.4 mg/ml of RNAP on R1.2/1.3 UltrAufoil grids was used. This allowed for a monodisperse distribution of particles with uniform ice thickness and with most squares in the grid that were of imageable quality when viewed at low magnification. We found that grids prepared with 0.1–0.2 mg/ml gave a good distribution of non-overlapping particles and proceeded to collect data with these grids for the 60° dataset. The behavior of particles at lower concentrations was different than the concentration of 0.4 mg/ml that was used for untilted data collection. Many squares were not imageable due to empty holes with no vitreous ice. Carefully selecting imageable areas over two grids enabled collection of a comparable amount of particles to that of untilted data collection scheme. For the 51° dataset, a grid prepared with 0.3 mg/ml of RNAP was used, which was less than what was used for untilted data collection but more than what was

used for 60° tilted collection. At this concentration, many squares were imageable, although we observed a more pronounced effect of squares having empty holes compared to the grid used for untilted data collection.

For the untilted dataset, a total of 638 micrographs was collected. After selecting micrographs with a CTF fit limit of under 6 Å, 619 micrographs were left with approximately 280 particles/micrograph. With this strategy, a sub-4 Å map was obtained using the standard data processing pipeline of cryoSPARC Live[33,50]. For the 60° dataset, a total of 826 micrographs was collected. After selecting micrographs with a CTF fit limit of under 6 Å, 527 micrographs were left with approximately 137 particles/micrograph. For the 51° dataset, a total of 1232 micrographs were collected. After selecting micrographs with a CTF fit limit of under 6 Å, 1024 micrographs were left with approximately 786 particles/micrograph.

Movies were initially aligned using UCSF MotionCor2[43] using 5 by 5 patches. Initial CTF estimates were derived using patch-based CTF estimation in cryoSPARC[50]. Particles were picked using blob picker in cryoSPARC, followed by one round of 2D classification. Particle class averages with high resolution features were used as templates for template-based particle picking in cryoSPARC. Particles were extracted from the micrographs and subjected to iterative 2D classification to obtain a clean stack of particles. GCTF[32] was then applied to obtain per-particle defocus values, which we found in our previous report yielded the highest-resolution reconstructions of specimens collected with the stage tilted[15], in comparison to CTFFind3[49] and CTFTilt[49]. An equal number of particles were then extracted from 0° and 60° tilt angle, in a manner that ensures that the spread of defocus values is similar across the different datasets (Supplementary Table 6), resulting in 72,000 particles per tilt angle.

Thereafter, 3D reconstructions were derived using cryoSPARC homogenous refinement[50], cryoSPARC CTF refinement[50] and RELION Bayesian polishing[52] (using v/d/a of 1/7000/0.5) until no further improvements in global resolution were observed. For processing in RELION, particle metadata STAR files were generated from cryoSPARC homogeneous refinement metadata files using pyem module and cspar2star.py script[69].

For 51° tilted dataset, similar data-processing steps were employed, except three 3D classes were obtained from one round of heterogeneous refinement from a clean stack of particles. Euler angles from these 3 classes were used for computing SCF[13] and the mean of 3 SCF values was reported.

For the map-to-model analysis, a refined atomic map of RNAP was used (deposited as PDB ID: 8TXO). The PDB was converted to a map using molmap function in UCSF Chimera[70]. The mask was generated using this PDB initial model which was binarized at 0.01 threshold, extended by 3 pixels, given a soft Gaussian edge of 3 pixels using 'relion_mask_create'. The sharpened proteasome maps were then compared against the aligned PDB generated maps, using the PDB generated mask, to obtain the map-to-model FSC. RELION was used to generate these FSC curves. SCF values were generated using the SCF command line tool[13,14], while 3DFSC results were generated using the 3DFSC web server[15]. Molecular graphics were generated using UCSF Chimera[70].

## Cryo-electron tomography data acquisition

Tomograms were collected for an RNA polymerase (0.3 mg/ml) grid using a Thermo Fisher Titan Krios TEM operating at 300 keV at the Pacific Northwest Center for Cryo-EM (PNCC). Gatan K3 BioQuantum direct electron detector equipped with an energy filter operated in zero-loss mode (with slit width of 20 eV) was used. The tomography data were acquired from specific regions of the grid that were not previously exposed and were similar in vitrification quality to the squares used for single-particle data collection. The software SerialEM[73] was utilized for data collection, employing a dose-symmetric scheme starting at 0° and

capturing alternating negative and positive tilts in 3° increments. The tilt angles for the tomograms ranged from −48° to +54°, while the nominal defocus range during tilt series collection was −2 to −4 μm. Each tilt was recorded as movie frames with 10 sub-frames in counting mode, utilizing a per-frame dose of approximately 0.23 e/Å$^2$. The magnification was set to 42Kx, resulting in a pixel size of 1.06 Å/pixel. A total of 25 tomograms were collected for the RNA polymerase sample, and among them, 5 with best alignment statistics were subjected to further processing and data analysis.

### Cryo-electron tomography data processing

Image stacks were subjected to frame-alignment and CTF estimation in Warp[75]. The frame-aligned averages were then exported to IMOD[76] for further processing. The Etomo module in IMOD was utilized for tomogram alignment using patch tracking and subsequent tomogram reconstruction. Prior to reconstruction, the aligned stack was binned by a factor of six (resulting in a pixel size of 6.36 Å) and a standard Gaussian filter was applied. Particles within the tomogram exhibited a tilt perpendicular to the field of view, approximately 4° away from the X-axis. For further validation, we extracted 25 micrographs at 0° tilt and performed CTF and tilt angle estimation in cryoSPARC using the Manually Curate Exposures function based on the defocus gradient[50] The observed tilt in the tomograms can be attributed either to a residual tilt angles present on the microscope stage or to a deformation of the grid caused during vitrification. To address this residual tilt, the coarse alignment and reconstruction were then recalculated by applying a −4° tilt to the x-axis. Tomograms were visualized in IMOD using the "3dmod Slicer" view.

Ice-thickness measurement was performed by employing a previously described method based on template matching and particle picking[77]. The positions of the top-most and bottom-most particles at the air-water interface were then used to delineate the ice boundary. To generate the reference model and mask required for template matching, the final RNA polymerase map obtained from the 60° tilted single particle analysis data collection. This map was low-pass filtered to 20 Å, resized, and resampled to match the pixel size of the tomogram (6.36 Å pixel size and a voxel size of 48). The contrast of the map was inverted to serve as the reference model. The same map was used to create the soft-edge mask. The processing steps, including low-pass filtering, contrast inversion, map down-sampling, and mask generation, were carried out using the e2proc3d.py script in EMAN2[78].

For particle picking, an additional mask was generated in the Amira software (Thermo Fisher Scientific) to exclude the gold edges of the grid in order to avoid false positives during template matching. Template matching and particle picking were performed using PyTom[79]. The resulting particle positions and initial orientations were then converted to the RELION 3.1 STAR file format[51]. To visualize the position of each particle within the tomogram, the ArtiaX[80] plugin in ChimeraX[81] was utilized (Fig. 7 and Supplementary Movie 1). The ice-thickness measurement was conducted using ChimeraX by determining the distance between the particles located at the air-water interface.

### Calculation of SCF for predicting optimal stage tilt angles

The construction of Fig. 10 involves the evaluation of the sampling compensation factor (SCF*) introduced in Baldwin and Lyumkis[13,14]. All of the distributions represented in Fig. 9 are shown in detail in this previous work. To obtain the values of the SCF, the graphical user interface (GUI) described in Baldwin and Lyumkis[14] was used after loading the 3 column files representing the Euler angles for 10,000 projections of the described distribution. A color map was chosen to effectively demonstrate the transition region from good to poor distributions. We take this transition value of the SCF to be 0.81, which is the theoretical and empirical value of the SCF for pure side views and describes a fully sampled reconstruction. A set of side view projections

fill the entirety of Fourier space, but show some variation in SNR, with accentuated SNR along the symmetry axis and lower values away from this axis.

### Reporting summary

Further information on research design is available in the Nature Portfolio Reporting Summary linked to this article.

## Data availability

All raw movie frames, micrographs, the particle stack, and relevant metadata files for the untilted and tilted datasets, generated in this study, are deposited in the EMPIAR database as EMPIAR-11791 (AAV2), EMPIAR-11792 (Apoferritin), EMPIAR-11796 (DPS) and EMPIAR-11797 (RNAP). The electron potential maps of AAV2 at various tilts, generated in this study, are deposited into the electron microscopy databank as EMD-36766 (0°), EMD-36767 (10°), EMD-36768 (20°), EMD-36769 (30°), EMD-36770 (40°), EMD-36771 (50°) and EMD-36772 (60°). The electron potential maps of apoferritin (Scripps Research) at various tilts, generated in this study, are deposited into the electron microscopy databank as EMD-36807 (0°), EMD-36809 (10°), EMD-36810 (20°), EMD-36811 (30°), EMD-36812 (40°), EMD-36814 (50°) and EMD-36813 (60°). The electron potential maps of apoferritin (Janelia Research Campus) at various tilts, generated in this study are deposited into the electron microscopy databank as EMD-41230 (0°) and EMD-41231 (30°). The electron potential maps of DPS at various tilts, generated in this study, are deposited into the electron microscopy databank as EMD-36816 (0°), EMD-36817 (10°), EMD-36818 (20°), EMD-36819 (30°), EMD-36820 (40°), EMD-36821 (50°) and EMD-36822 (60°). The model for RNAP, generated in this study, has been deposited into the PDB as PDB ID-8TXO and the electron potential map for the 60° tilted dataset, generated in this study, is deposited into the electron microscopy databank as EMD-41695. Other data are available from the corresponding authors upon request. PDB codes of previously published structures used in this study are 6E9D (AAV2), 7A6A (apoferritin), 1YAR (archeabacterial 20S proteasome), and 6GCM (DPS). Source data are provided with this paper.

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

## Acknowledgements

We would like to thank the late Dr. Mavis Agbandje-McKenna for her pioneering studies of parvovirus capsid structures. We thank Chris Russo for the plasmid for DPS, and Yifan Cheng for the 20S proteasome sample. We thank Nikolaus Grigorieff, Timothy Grant, Alexis Rohou, Anchi Cheng, Alex Noble, Bridget Carragher, and Clinton S. Potter for their invaluable advice. We thank Bill Anderson at TSRI for help with microscope maintenance, Laura Yen at NYSBC for assistance with freezing grids and single-particle data collection of the 20S proteasome, and Craig K. Yoshioka at PNCC for assistance with tomography data collection. We thank the Center for Bioimaging Sciences (CBIS) NUS Singapore for computation and cryo-EM access. Molecular graphics and analyses performed with UCSF ChimeraX, developed by the Resource for Biocomputing, Visualization, and Informatics at the University of California, San Francisco, with support from National Institutes of Health R01-GM129325 and the Office of Cyber Infrastructure and Computational Biology, National Institute of Allergy and Infectious Diseases. The work was supported by National Institutes of Health (U54 AI170855 and U01 AI136680 to D.L., GM148476 to D.W., GM082946 to R.M. and M.A.M.) the National Science Foundation (MCB 2048095 to D.L.); Hearst Foundations Developmental Chair and the Margaret T. Morris Foundation (to D.L.); National Research Foundation, Singapore (NRF Fellowship A-8001346-00-00 to Y.Z.T.); National University of Singapore (PYP Fellowship A-0008405-00-00, A-0008405-01-00 to Y.Z.T.); Ministry of Education, Singapore (MOE AcRF Tier 1 A-8000037-00-00, MOE AcRF Tier 2 A-6100427-01-00 to Y.Z.T.); Agency for Science, Technology and Research, Singapore (to Y.Z.T.). D.A.G. is supported by the Nadia's Gift Foundation Innovator Award of the Damon Runyon Cancer Foundation (DRR-65-21). B.A.B. is supported by an American Cancer Society postdoctoral fellowship (PF-21-075-01-CCB). D.L. also acknowledges support from the Cancer Center Support Grant P30 CA01495. Some of the work was performed at the National Resource for Automated Molecular Microscopy at the Simons Electron Microscopy Center which is supported by National Institute of General Medical Sciences (GM103310), Simons Foundation (SF349247), and NYSTAR. The cryo-EM data was collected at Scripps Research (supported by S10 OD032467), New York Structural Biology Center, and Janelia Research Campus. S.H. and S.A.B., and the development of unnatural base pairs were supported by the National Science Foundation under Grant MCB-1939086. Any opinions, findings and conclusions or recommendations expressed in this material are those of the author(s) and do not reflect the views of National Research Foundation, Singapore; National University of Singapore; Ministry of Education, Singapore; and Agency for Science, Technology and Research, Singapore.

## Author contributions

D.L. and Y.Z.T. conceived the project. M.M. and J.A.H. purified AAV2 under supervision of R.M. and M.E.B. and G.L. expressed and purified DPS under J.A.P.N.'s supervision. J.O. purified RNAP under D.W.'s supervision. S.H. made the artificial bases under S.A.B.'s supervision. S.Đ.M. and S.A. prepared apoferritin samples for data collection at Scripps Research. D.P. prepared apoferritin samples for data collection at Janelia Research Campus and S.A. collected and processed the cryo-EM data collected at Scripps Research and Janelia Research Campus. S.A. froze the grids for DPS, and Y.Z.T. collected the cryo-EM data. S.A. froze the RNAP grids and collected the cryo-EM data. Y.Z.T. collected the proteasome cryo-EM data. S.A., S.M.T., and Y.Z.T. processed the cryo-EM data. Z.S. and J.O. built and refined the model for RNAP. P.R.B. performed the SCF calculations. A.M. collected, processed, and analyzed RNAP tomography data with B.A.B., D.A.G., and D.L. providing guidance and supervision. R.M., M.A.M., J.A.P.N., D.W. and D.L. provided funding support. S.A., P.R.B., Y.Z.T. and D.L. wrote the manuscript, with input from all co-authors.

## Competing interests

The authors declare no competing interests.

## Additional information

[1]Laboratory of Genetics, The Salk Institute for Biological Studies, La Jolla, CA 92037, USA. [2]Department of Biochemistry and Molecular Biology, Baylor College of Medicine, Houston, TX 77030, USA. [3]Department of Biological Sciences, National University of Singapore, 16 Science Drive 4, Singapore 117558, Singapore. [4]Division of Pharmaceutical Sciences, Skaggs School of Pharmacy & Pharmaceutical Sciences, University of California, San Diego, La Jolla, CA 92093, USA. [5]College of Pharmacy, Kyung Hee University, Seoul 02247, Republic of Korea. [6]Jack H. Skirball Center for Chemical Biology and Proteomics, The Salk Institute for Biological Studies, La Jolla, CA 92037, USA. [7]Department of Biochemistry and Molecular Biology, College of Medicine, University of Florida, Gainesville, FL 32610, USA. [8]Foundation for Applied Molecular Evolution, 13709 Progress Blvd Box 7, Alachua, FL 32615, USA. [9]Department of Integrative Structural and Computational Biology, The Scripps Research Institute, La Jolla, CA 92037, USA. [10]Department of Chemistry and Biochemistry, University of California San Diego, La Jolla, CA 92093, USA. [11]Department of Cellular and Molecular Medicine, University of California, San Diego, La Jolla, CA 92093, USA. [12]Disease Intervention Technology Laboratory (DITL), Agency for Science, Technology and Research (A*STAR), 8A Biomedical Grove, Singapore 138648, Singapore. [13]Institute of Molecular and Cell Biology (IMCB), Agency for Science, Technology and Research (A*STAR), 61 Biopolis Drive, Proteos, Singapore 138673, Republic of Singapore. [14]Graduate School of Biological Sciences, Section of Molecular Biology, University of California San Diego, La Jolla, CA 92093, USA. [15]Deceased: Mavis Agbandje-McKenna. ✉e-mail: yztan@nus.edu.sg; dlyumkis@salk.edu

