## [Peer Review File · Nature Communications]

Overcoming Resolution Attenuation During Tilted Cryo-EM Data CollectionREVIEWER COMMENTS

Reviewer #1 (Remarks to the Author):

This manuscript by Aiyer et al. describes the analyses of the effects of tilting the stage on single-particle cryo-EM reconstructions using multiple specimens. This is a thorough study that addresses a major problem in cryo-EM single particle analysis – sample preferred orientation. They reveal that, after higher-order data processing steps, resolution of data collected at up to 60 degree tilt can approach that collected on the same sample untilted but with better sampling. Such analysis is long-overdue and exciting to see(!), and we can now use it to assure users (and skeptics) that collecting tilted data is a very viable approach to overcoming preferred orientation and that it should be tried rather than continuously testing new freezing conditions and wasting valuable sample to correct the orientation problem at the sample prep stage. The manuscript is clearly written and easy to follow. In addition to being valuable to a general cryo-EM user audience, it will also be useful as a teaching tool for cryo-EM novices. I have only a few comments to be addressed:

- Have the authors noticed large differences in resolution improvements after CTF refinement in CryoSPARC as compared to Relion? The apoferritin compared to AAV2 analyses suggest there may be. Please comment on this.
- Figure 5 does not line up with description in text – please fix.
- I am very curious as to what effect merging tilted and untilted data would have on sampling, resolution, and map quality as compared to the improvements shown in the presented workflow. Would this lead to better sampling and a resolution that is an average of what was achieved with the untilted and tilted datasets, or would it make an even better map achievable? Please try your data processing workflow after merging particle stacks from the tilted and untilted data sets from RNAP and include this analysis. This would be very useful information for those who collect tilted data.

Reviewer #2 (Remarks to the Author):

Peer review of manuscript NCOMMS-23-31730: “Overcoming Resolution Attenuation During Tilted Cryo-EM Data Collection”.

The study by Aiyer et al. examines in detail the primary strategy (tilt acquisition) for generic processing of cryo-EM single particle data acquisition for samples suffering from severe orientation preference issues. Validation of tilt acquisition strategies and their workflow using higher symmetry subjects demonstrates what one would expect with higher symmetry, using the de facto standard of (apo)ferritin, along with a small virus (AAV2) approximately two and a half times the size of (apo)ferritin and a lower symmetry smaller complex approximately half the size of (apo)ferritin. This is to provide a uniform analysis of the impact of tilt acquisition. Finally, using an asymmetric ribosome which demonstrates severe orientation

preference (to the point where reconstructions are essentially unusable without improving variance in sampling angles) they demonstrate that their strategy can be applied with great success. Finally, they briefly examine what degree of tilt is required to compensate for a given severity of orientation preference.

This work is thorough, and uses validation metrics (3D FSC, sampling compensation factor) which the authors have developed previously. It replicates many things already well understood in cryo-EM, and already public individually (tilt acquisition to overcome orientation preference (Tan et al., 2017)), optical parameter optimisation (Brown et al., 2022; Nakane et al., 2020), cycling through multiple rounds of optical parameter and motion refinement (Moriya et al., 2020). It does show what can be achieved using a combination of modern data acquisition and processing methods very well.

The estimation of required tilt angle necessary to overcome a particular degree of orientation preference is of particular interest in the future.

Major concerns:

No major concerns.

Minor concerns:

The most novel part of this manuscript, that of defining requisite tilt angles to overcome a given orientation preference, receives less treatment in the text than I think it deserves. Could the minimum tilt required for sufficient sampling of the RNAP (for example) be calculated given the orientation distribution (with varying "sprinkles") just as an addition?

I would have liked to have seen results for a second sample demonstrating severe orientation preference. As Fig. 10 demonstrates two types of sampling, perhaps a C2 symmetry sample which demonstrates a strong preferred orientation would have been interesting. I acknowledge at this stage this is unlikely to be possible due to additional time cost.

The methods are mostly exhaustively detailed. However, with respect to Bayesian polishing, was training carried out to optimise parameters (sigma for velocity, divergence, and acceleration) for each dataset or not? If so, how many particles were used for training? If not, were the default parameters in RELION used? If the defaults were not used, what parameters were used? Were the parameters changed for subsequent rounds of Bayesian polishing? Did the B-factor estimation curves change measurably between the first and subsequent rounds of Bayesian polishing? If so, by how much?

With Ewald sphere correction, was it also tested in RELION, as it uses a different implementation from CryoSPARC?

Text is ambiguous regarding a third round of Bayesian polishing: Ln. 239-240 implies only a third round of

CTF refinement (no third round of Bayesian polishing), while Ln. 275 implies pairing CTF refinement and Bayesian polishing (thus Bayesian polishing carried out three times).

Gas mix used for plasma treatment of grids is described for proteasome (Ln. 942) but not specified for AAV2 (Ln. 797-799) and apoferritin (Ln. 855-856). Were they the same?

Naming/capitalisation of direct detectors in prose; Lns. 808, 867, 906, 950, 1013. Is the K3 Quantum the BioQuantum?

Two citations missing (placeholder text present) on Lns. 915 and 916.

Is accuracy to five decimal places necessary for image acquisition (Ln. 950)?

Formatting for *E. coli*, Lns. 985 and 988.

Unnecessary punctuation in Ln. 1037.

Ln. 1069, 1071, 1073 – acronym EB not explained, presumed to be ‘elution buffer’?

EMAN was used for map-to-model FSC calculations for AAV2 (Ln. 842), but RELION was used for DPS (Ln. 1044)? Were they processed by different people?

RNAP cryo-EM data collection prose is missing information regarding microscope and detector used. This is present in details for other samples (Ln. 1082).

Number of particles per micrograph for the 51° RNAP appears to be an order of magnitude too large given prior text and information in Supplementary Table 6 (Ln. 1103).

Ln. 1139-1140: The K3 Summit does not exist; should it be K3 (Bio)Quantum given Supplementary Table 6?

Typographic error in Ln. 1200 (‘Apoferritn’).

EMDB access codes for RNAP not listed in ‘Accession code and deposition’ information. Listed as ‘X’ in Supplementary Table 6.

Please check grammar throughout, e.g.: unnecessary usage of ‘the’ at Ln. 906, incorrect tense in Ln. 1017, incorrect usage of ‘the’ at Ln. 1140, unnecessary usage of ‘the’ at Ln. 1143, ‘the’ missing twice at Ln. 1149.

Citation concerns:

Reference 18 appears to contain an extra carriage return (Ln. 1249-1251).

Reference 66 appears to be broken, containing both a citation for PyEM and for a J. Mol. Biol. chromatin manuscript (Ln. 1350-1353).

References for above:

Brown, Z. P., Abaeva, I. S., De, S., Hellen, C. U. T., Pestova, T. V., & Frank, J. (2022). Molecular architecture of 40S translation initiation complexes on the hepatitis C virus IRES. *EMBO J*, 41(16), e110581.

<https://doi.org/10.15252/embj.2022110581>

Moriya, T., Adachi, N., Kawasaki, M., Yamada, Y., Shinoda, A., Koiwai, K., Yumoto, F., & Senda, T. (2020). Size matters: optimal mask diameter and box size for single-particle cryogenic electron microscopy.

bioRxiv, 2020.2008.2023.263707. <https://doi.org/10.1101/2020.08.23.263707>

Nakane, T., Kotecha, A., Sente, A., McMullan, G., Masiulis, S., Brown, P., Grigoras, I. T., Malinauskaite, L., Malinauskas, T., Miehl, J., Uchanski, T., Yu, L., Karia, D., Pechnikova, E. V., de Jong, E., Keizer, J., Bischoff, M., McCormack, J., Tiemeijer, P., . . . Scheres, S. H. W. (2020). Single-particle cryo-EM at atomic resolution. *Nature*, 587(7832), 152-156. <https://doi.org/10.1038/s41586-020-2829-0>

Tan, Y. Z., Baldwin, P. R., Davis, J. H., Williamson, J. R., Potter, C. S., Carragher, B., & Lyumkis, D. (2017). Addressing preferred specimen orientation in single-particle cryo-EM through tilting. *Nat. Methods*, 14(8), 793-796. <https://doi.org/10.1038/nmeth.4347>

Reviewer #3 (Remarks to the Author):

Sriram Aiyer et al reported overcoming resolution attenuation using tilted cryo-EM data collection. The authors started with highly symmetrical particles with relatively large molecular weight to demonstrate that high-resolution information can be reliably recovered through an optimized data collection and processing strategy. They then showed for relatively “small” particles with D or O symmetry, the high-resolution structure can still be achieved. Mostly importantly, for a dataset showing strong-preferred orientation with C1 symmetry, which represents an authentic scenario where tilted data collection is required, convincingly demonstrate the effectiveness of the data processing pipeline. Furthermore, the authors present a quantitative framework to allows researchers to define an optimal tilt angle for data acquisition. Overall, the work presented here is sound. The methodology and data analysis are well-documented and well-reasoned, which is greatly appreciated by the reviewer.

Major question:

The authors first demonstrate that resolution attenuation is minimal for large protein complexes at highly tilted images, which is convincing. However, for many projects which work on relatively small molecules (around 100kDa to 150 KDa), it is generally more difficult to achieve high resolution. The authors have discussed the difficulty for small protein complexes and possible solutions briefly in Line517-524. The question is, will tilted data collection help resolve the preferred-orientation issue for

some really challenging samples like some of the membrane proteins in real scenario? Although it has been shown that tilted stage improved density maps for very small proteins at ~43 Kda (Herzik et al Nature Comm 2019), the final resolution is still limited. To strengthen the claim that “resolution attenuation is negligible or significantly across tilt angles”, the authors should demonstrate whether a relatively small molecule (within 200kda, no symmetry) can benefit from the tilted data collection, given the obvious drop in relative SNR of the particles. The drawbacks of tilted data collection should be discussed and highlighted.

Minors

1. For each dataset, a few rounds of 2D classification were performed to generate equal amount of dataset for comparison, which is methodologically sound. Could the author plot the particle location distribution in the original micrographs? Especially for the high-tilt images, are the particles along the tilt-axis better than particles close to the micrograph edges?
2. Line 323: “Importantly, our results show that with appropriate data processing, tilting the stage is a viable strategy to obtain high-resolution reconstructions even for small proteins suffering from preferred orientation.” DPS dataset does not suffer from preferred orientation issue, therefore this statement is not valid. Also a 230kda complex with tetrahedral symmetry is not a small protein complex.
3. L543-544, The actual ice layer tilt angle could be slightly different to the tilt angle. Could the authors provide the calculated tilt angle distribution for each tilted dataset? This is provided from cryoSPARC.
4. Are the microscopes used in this study single-tilt stage or double-tilt stage?
5. In general, a relatively small dataset was collected from each sample. For example, in Line910-911, “A total of 637 micrographs was recorded over a single 4-day collection using stage shift, across tilt angles of 0° and 30° set with stage alpha.” This is considerably lower throughput compared with modern capacity of Krios microscope. Could the author provide more information regarding the speed of data collection? Or is a longer waiting time is setup for stage drift? Any difference in terms of stage settling time at different stage tilt? These information will be useful for readers to transfer the knowledge of this study to other applications.

We would like to thank the reviewers for their critical evaluation and careful reading of our manuscript, and their overall support of the work. Their suggestions and comments have allowed us to clarify the text / figures and strengthen the data. In addition to making minor modifications to the text, we have performed several new experiments, which added several supplementary figures, as well as a Supplementary Note. Supplementary Figures 6-7, along with Supplementary Note 2, have been added to show that there are no added benefits to combining tilted and untilted data in the experiment. Supplementary Figures 8-10 indicate that the particles that contributed to the final reconstructions do not reside along the tilt axis, and are in fact randomly distributed throughout the micrographs. The reviewers' specific concerns are addressed point-by-point below.

REVIEWER COMMENTS

Reviewer #1 (Remarks to the Author):

This manuscript by Aiyer et al. describes the analyses of the effects of tilting the stage on single-particle cryo-EM reconstructions using multiple specimens. This is a thorough study that addresses a major problem in cryo-EM single particle analysis – sample preferred orientation. They reveal that, after higher-order data processing steps, resolution of data collected at up to 60 degree tilt can approach that collected on the same sample untilted but with better sampling. Such analysis is long-overdue and exciting to see(!), and we can now use it to assure users (and skeptics) that collecting tilted data is a very viable approach to overcoming preferred orientation and that it should be tried rather than continuously testing new freezing conditions and wasting valuable sample to correct the orientation problem at the sample prep stage. The manuscript is clearly written and easy to follow. In addition to being valuable to a general cryo-EM user audience, it will also be useful as a teaching tool for cryo-EM novices. I have only a few comments to be addressed:

Response: We thank the reviewer for their positive feedback to our work.

- Have the authors noticed large differences in resolution improvements after CTF refinement in CryoSPARC as compared to Relion? The apoferritin compared to AAV2 analyses suggest there may be. Please comment on this.

Response: We did not observe any difference in resolution improvement when CTF refinement was performed in cryoSPARC versus RELION for the AAV2 dataset. Hence, for apoferritin and DPS, we used the cryoSPARC implementation, because it was simpler to run immediately after refinement and did not require additional conversion steps. We now add a line in the Methods section to clarify this (Page 34, Lines 934-939).

- Figure 5 does not line up with description in text – please fix.

Response: Fixed. Renumbered to accurately follow what is described in the figure. Panel “g” was erroneously included, and this has been removed (Page 10, Lines 319-320).

- I am very curious as to what effect merging tilted and untilted data would have on sampling, resolution, and map quality as compared to the improvements shown in the presented workflow. Would this lead to better sampling and a resolution that is an average of what was achieved with the untilted and tilted datasets, or would it make an even better map achievable? Please try your data processing workflow after merging particle stacks from the tilted and untilted data sets from RNAP and include this analysis. This would be very useful information for those who collect tilted data.

Response: This is one of the most frequently asked questions with regards to tilting. In Figure 2 of Tan *et al.*¹, using the HA trimer, we previously showed that the true quality of the map, and its resolution (as assessed using the map-to-model metric, i.e. using an external positive control reference), from a dataset combined from tilted and untilted images was inferior to the quality and resolution of a map from tilted images alone. Based on these data, and supported by theoretical calculations reported later (Lyumkis and Baldwin^{2,3}), our recommendation has always been to not combine tilted and untilted data, because the untilted images hyper-sample around a single Fourier plane, which would not be expected to yield constructive improvements. Now, we have obtained data using RNAP, which is better resolved than the previous HA data and does not suffer from problems with overfitting (reported on previously in Tan *et al.*¹). With the new data in hand, we have again performed the suggested experiment of combining untilted and tilted data. In brief, our primary conclusion remains the same — that adding untilted images does *not* improve any of the quantitative validation metrics nor the quality of the map.

We wished to gain further insights from the new dataset, and we uncovered some interesting nuances, as described below, from two separate experiments.

- (1) We first merged 36,000 randomly selected untilted particles with 36,000 randomly selected tilted particles, yielding a 72,000-particle stack that could be directly compared to refinements from either untilted or tilted data alone. We found that the global resolution and the 3DFSC is between that of untilted and the tilted dataset. The SCF of the merged stack approached the SCF of the tilted stack alone. We attribute this to the fact that the reconstruction is already well-sampled, owing to the presence of tilted images. We then merged the complete particle stacks of untilted and tilted RNAP datasets (72,000 + 72,000 particles = 144,000 particles) and found that the global resolution, the 3D FSC, and the quality of the map was almost identical to the resolution of reconstructions from the tilted images. In other words, the combination of these two stacks, doubling the number of particles, still does not yield better results compared to collecting and processing 60° tilted images alone. This is because the untilted particles do not improve the SCF and largely sample around a single Fourier plane, thereby minimally contributing to further improving the map.
- (2) From the entire 144,000-particle stack, we then incrementally removed the tilted particles – 100 particles at a time – and computed the SCF. As expected, when we removed the tilted particles, eventually leaving only the untilted particles contributing to the reconstruction, the SCF systematically decreases and approaches that of the untilted

stack, which is SCF=0.45. This experiment served as a positive control and again indicated that it is the tilted particles that contribute to the isotropic, high-quality map. Intriguingly, when we then performed the converse experiment, removing particles from the untilted stack, but leaving the tilted particles in the reconstruction, we noticed that the SCF actually gradually increases to ~ 0.83 , but then starts to drop back to its baseline value of ~ 0.78 . In other words, there appeared to be a narrow window within which combining some small amount of untilted images with primarily tilted images may make sense, based solely on the SCF value. However, when we then refined individual stacks containing varying amounts of untilted images with primarily tilted images, *i.e.* in accordance with the prediction from SCF, we again did not notice any tangible improvements in comparison to reconstructions from the tilted images alone. The best explanation for this is based on the interpretation of the SCF, which is directly and linearly related to the spectral signal-to-noise ratio (SSNR). We know that $SSNR = FSC / (1 - FSC)$, and by extension, $FSC = SSNR / (1 + SSNR)$. A small improvement in SSNR (and by extension, the SCF), such as from $0.79 \rightarrow 0.85$, will yield an even smaller improvement in the FSC. Although we do observe statistically significant improvements to the 3DFSC, as may have been predicted, this is likely because the fluctuations of the FSC curves are within the noise limits of detection. We conclude, as previously, that there is no current evidence to suggest that combining tilted and untilted images should yield clear-cut benefits to the experimentalist; however, we cannot exclude the possibility outright that some small improvements could be obtained, for example when the orientation distribution is less pathological, or when a small number of untilted images are included, e.g. to not overwhelm the sampling distribution.

These data are of sufficient interest that we now report them in **Supplementary Figures 6-7**, and we include a discussion of the results in **Supplementary Note 2** (referenced in text on Page 12, Lines 394-396). We thank the reviewer for this valuable suggestion.

References

- ¹Tan, Y.Z., Baldwin, P.R., Davis, J.H., Williamson, J.R., Potter, C.S., Carragher, B. and Lyumkis, D., 2017. Addressing preferred specimen orientation in single-particle cryo-EM through tilting. *Nature methods*, 14(8), pp.793-796.
- ²Baldwin, P.R. and Lyumkis, D., 2020. Non-uniformity of projection distributions attenuates resolution in Cryo-EM. *Progress in biophysics and molecular biology*, 150, pp.160-183.
- ³Baldwin, P.R. and Lyumkis, D., 2021. Tools for visualizing and analyzing Fourier space sampling in Cryo-EM. *Progress in biophysics and molecular biology*, 160, pp.53-65.

Reviewer #2 (Remarks to the Author):

Peer review of manuscript NCOMMS-23-31730: "Overcoming Resolution Attenuation During Tilted Cryo-EM Data Collection".

The study by Aiyer et al. examines in detail the primary strategy (tilt acquisition) for generic processing of cryo-EM single particle data acquisition for samples suffering from severe orientation preference issues. Validation of tilt acquisition strategies and their workflow using higher symmetry subjects demonstrates what one would expect with higher symmetry, using the de facto standard of (apo)ferritin, along with a small virus (AAV2) approximately two and a half times the size of (apo)ferritin and a lower symmetry smaller complex approximately half the size of (apo)ferritin. This is to provide a uniform analysis of the impact of tilt acquisition. Finally, using an asymmetric ribosome which demonstrates severe orientation preference (to the point where reconstructions are essentially unusable without improving variance in sampling angles) they demonstrate that their strategy can be applied with great success. Finally, they briefly examine what degree of tilt is required to compensate for a given severity of orientation preference.

This work is thorough, and uses validation metrics (3D FSC, sampling compensation factor) which the authors have developed previously. It replicates many things already well understood in cryo-EM, and already public individually (tilt acquisition to overcome orientation preference (Tan et al., 2017)), optical parameter optimisation (Brown et al., 2022; Nakane et al., 2020), cycling through multiple rounds of optical parameter and motion refinement (Moriya et al., 2020). It does show what can be achieved using a combination of modern data acquisition and processing methods very well.

The estimation of required tilt angle necessary to overcome a particular degree of orientation preference is of particular interest in the future.

Response: We thank the reviewer for the positive feedback.

Major concerns:

No major concerns.

Minor concerns:

The most novel part of this manuscript, that of defining requisite tilt angles to overcome a given orientation preference, receives less treatment in the text than I think it deserves. Could the minimum tilt required for sufficient sampling of the RNAP (for example) be calculated given the orientation distribution (with varying "sprinkles") just as an addition?

Response: Yes, this is precisely the argument we put forward in Figure 10. Given a set of orientation distributions alone, we can now quantitatively predict the optimal tilt angle required to obtain near-uniform or "side-like" sampling.

The software that we have distributed can take, as input, any type of orientation distribution of any number of particles to define the current SCF, and to predict the SCF when stage tilt is used. What we show in Fig 10 are just two examples of orientation distribution to visually illustrate how the SCF changes as a function of the angle of preferred orientation, the amount of “sprinkled” projections outside of the main view, and the stage tilt angle. An example of the utility of the SCF software is provided below. We could predict the optimal tilt angle for RNAP data collection using Euler angles obtained from a refinement using the untilted dataset during cryoSPARC live. The SCF value that we obtained was ~ 0.36 and by sliding the widget button, we predicted that 51° should be the optimal tilt angle to obtain uniform side-like views.

To further emphasize this point, we have added this sentence in the discussion: “This graphical representation visually demonstrates how the SCF changes when there are additional views or when stage tilt is applied and complements the previously described software to explicitly measure the SCF of tilted data given any orientation distribution¹.” (Page 16, Lines 550-552)

Below is an example snapshot of the SCFgui¹, where using initial set of particle projections an SCF value is computed and how introducing tilts can improve the SCF to ~ 0.81 for uniform side-like views.

References

¹Baldwin, P.R. and Lyumkis, D., 2021. Tools for visualizing and analyzing Fourier space sampling in Cryo-EM. *Progress in biophysics and molecular biology*, 160, pp.53-65.

I would have liked to have seen results for a second sample demonstrating severe orientation preference. As Fig. 10 demonstrates two types of sampling, perhaps a C2 symmetry sample which demonstrates a strong preferred orientation would have been interesting. I acknowledge at this stage this is unlikely to be possible due to additional time cost.

Response: We currently have a tilted cryo-EM dataset of a ~120 kDa protein complex, which is characterized by C2 rotational symmetry and maintains a pathologically preferential distribution due to adherence to the air/water interface. Using the strategies described in the manuscript, and tilts up to 50°, we were able to derive an accurate reconstruction of the complex assembly, resolved to ~3.5 Å. This data is shown below. Thus, we do not see any impediment to using our strategies for tilted single-particle cryo-EM data acquisition, even for small particles containing low symmetry, or for asymmetric particles for that matter. The ideas are generalizable. However, because this sample is of significant novel biological interest, and it represents the subject of a separate research direction, we cannot include these data in the current manuscript. We intend to report on these data within the next year. Please also see our response to reviewer #3 below.

The Euler angle distribution for the C2-symmetric object:

The corresponding map and atomic model:

The methods are mostly exhaustively detailed. However, with respect to Bayesian polishing, was training carried out to optimise parameters (sigma for velocity, divergence, and acceleration) for each dataset or not? If so, how many particles were used for training? If not, were the default parameters in RELION used? If the defaults were not used, what parameters were used? Were the parameters changed for subsequent rounds of Bayesian polishing? Did the B-factor estimation curves change measurably between the first and subsequent rounds of Bayesian polishing? If so, by how much?

Response: We did not do any training for Bayesian polishing. We followed the suggestions of Takanori Nakane (one of the Relion developers) on CCPEM. The only change that we make is that we consider the size of the particle, as clarified by Oliver Clarke in that same thread referenced above:

The parameters I use are:

For large (MDa) particles (v/d/a): 0.5/12000/1

For small (~150kDa) particles (v/d/a): 1/7000/0.5

Cheers

Oli

We now include the values in our methods for completeness (Page 34, Lines 929-930; Page 36, Lines 995-996; Page 37, Line 1037; Page 38, Line 1081; Page 40, Lines 1145-1146; Page 43, Line 1242).

With Ewald sphere correction, was it also tested in RELION, as it uses a different implementation from CryoSPARC?

Response: Ewald sphere curvature correction was applied using the implementation provided in cryoSPARC for the reported experiments. We did perform preliminary tests using the RELION implementation for the untilted dataset of AAV, but the resolution was not any different in comparison to the cryoSPARC implementation. Thus, we employed the cryoSPARC implementation for the rest of the work, which was simpler. Using the cryoSPARC implementation allowed us to perform the homogeneous refinement of the stack together with the Ewald sphere curvature correction in the same software.

Separately, for AAV2, we tried the RELION implementation in our 2018 Nature Communications paper¹ (Tan & Aiyer *et al.*), but it yielded similar results in comparison to the FREALIGN implementation. cryoSPARC's implementation follows the FREALIGN algorithm, and not the RELION one.

Because Ewald sphere curvature correction is not the focus of the current manuscript, we stick to the cryoSPARC implementation for ease of use.

References

¹Tan, Y.Z., Aiyer, S., Mietzsch, M., Hull, J.A., McKenna, R., Grieger, J., Samulski, R.J., Baker, T.S., Agbandje-McKenna, M. and Lyumkis, D., 2018. Sub-2 Å Ewald curvature corrected structure of an AAV2 capsid variant. *Nature communications*, 9(1), p.3628.

Text is ambiguous regarding a third round of Bayesian polishing: Ln. 239-240 implies only a third round of CTF refinement (no third round of Bayesian polishing), while Ln. 275 implies pairing CTF refinement and Bayesian polishing (thus Bayesian polishing carried out three times).

Response: Thank you for catching this. It is now fixed. The sentence now reads: "No further improvements were observed with the third round of CTF refinement for reconstructions from data collected at 20°-60°stage tilt" (Page 7 Line 240 to page 8 Lines 241-242). To reduce ambiguity, the sentence at Page 8 Lines 275-277 now reads "We observed further improvements in global resolution after the 2nd round of CTF refinement and Bayesian polishing (Fig. 3e-f), but not after the 3rd round (Fig. 3g)."

Separately, different rounds of CTF refinement were performed because for each dataset, iterative CTF refinement was done until no improvement was obtained. Hence, the actual number of rounds of CTF refinements may differ based on the dataset.

Gas mix used for plasma treatment of grids is described for proteasome (Ln. 942) but not specified for AAV2 (Ln. 797-799) and apoferritin (Ln. 855-856). Were they the same?

Response: Fixed. For AAV2 and apoferritin we used “75% argon/25% oxygen atmosphere at 15 Watts for 7 s”. We now include this on Page 33, Line 890 for AAV2; Page 35, Lines 963-964 for apoferritin; Page 39, Lines 1116-1117 for DPS; and Page 41, Lines 1193-1194 for RNAP).

Naming/capitalisation of direct detectors in prose; Lns. 808, 867, 906, 950, 1013. Is the K3 Quantum the BioQuantum?

Response: Fixed, K3 in the microscopes used for data collection is BioQuantum, except for AAV2 and proteasome which were collected using the K2 Summit. We have accordingly fixed the supplementary tables as well, which now indicate K3 BioQuantum.

Two citations missing (placeholder text present) on Lns. 915 and 916.

Response: Thank you, we have now fixed the first citation. There is no second citation. (Page 37, Line 1026-1028)

Is accuracy to five decimal places necessary for image acquisition (Ln. 950)?

Response: Fixed, now reads 1.07 (Page 38, Line 1062).

Formatting for E. coli, Lns. 985 and 988.

Response: Fixed, throughout the manuscript it now reads “E. Coli RNAP” (Page 41, Line 1186).

Unnecessary punctuation in Ln. 1037.

Response: Removed the comma (Page 40, Line 1150).

Ln. 1069, 1071, 1073 – acronym EB not explained, presumed to be ‘elution buffer’?

Response: Fixed, references to EB (elongation buffer) in the manuscript now expanded (Page 41, Line 1185).

EMAN was used for map-to-model FSC calculations for AAV2 (Ln. 842), but RELION was used for DPS (Ln. 1044)? Were they processed by different people?

Response: YZT performed both map-to-model calculations. He used EMAN 1.9 initially because it was installed on an older workstation, but after upgrading to a newer Linux workstation, he was not able to install EMAN 1.9 due to missing dependencies, and after numerous attempts he gave up and used RELION instead. FSC calculations are independent of software, hence there should not be any technical issues.

RNAP cryo-EM data collection prose is missing information regarding microscope and detector used. This is present in details for other samples (Ln. 1082).

Response: Fixed. We have added the following sentences on page 42, lines 1201-1204: “Images were recorded on a Titan Krios electron microscope (FEI/Thermo Fisher) equipped with a K3 BioQuantum direct detector (Gatan) at 0.83 Å per pixel in counting mode (CDS enabled) using the Leginon software package. We used an energy filter slit width of 20 eV. Movies were recorded using a beam-image shift targeting strategy with the 100 µm objective aperture inserted.”

Number of particles per micrograph for the 51° RNAP appears to be an order of magnitude too large given prior text and information in Supplementary Table 6 (Ln. 1103).

Response: The quality of the 51°-tilted dataset is not as good as the quality of the 0° dataset or the 60° dataset. The grid that we used for the 51° was more crowded, and during data collection, we were also picking up more ice edges and false positive picks that do not contribute to the final reconstruction. This could be for a large variety of reasons. As a result, we needed to perform extensive cleaning for *this* dataset, including iterative 2D and 3D classification, in order to obtain a good quality stack. We note that we did not experience this same problem for the 60°-dataset, indicating that the issue was inherent to the sample and/or grid preparation, not to the fact that we were collecting tilted data. We do, however, note that, when tilts are employed, more attention needs to be placed on the particle distribution, which we commented upon in the original submission, and now we elaborate upon in the discussion on Page 14, Line 477-484 and the Methods on page 42 lines 1212-1219.

Ln. 1139-1140: The K3 Summit does not exist; should it be K3 (Bio)Quantum given Supplementary Table 6?

Response: Fixed. We used K3 BioQuantum.

Typographic error in Ln. 1200 (‘Apoferritn’).

Response: Fixed (Page 46, Line 1324).

EMDB access codes for RNAP not listed in ‘Accession code and deposition’ information. Listed as ‘X’ in Supplementary Table 6.

Response: Fixed. The EMDB accession code 41695 and PDB code 8TXO have been added in **Supplementary Table 6** and in the ‘Access code and deposition’ information section.

Please check grammar throughout, e.g.: unnecessary usage of ‘the’ at Ln. 906, incorrect tense in Ln. 1017, incorrect usage of ‘the’ at Ln. 1140, unnecessary usage of ‘the’ at Ln. 1143, ‘the’ missing twice at Ln. 1149.

Response: Thanks, fixed. We also went through the entire manuscript again and made changes to grammar throughout.

Citation concerns:

Reference 18 appears to contain an extra carriage return (Ln. 1249-1251).

Response: Thanks, removed the extra carriage return.

Reference 66 appears to be broken, containing both a citation for PyEM and for a J. Mol. Biol. chromatin manuscript (Ln. 1350-1353).

Response: Thanks, fixed the error.

Reviewer #3 (Remarks to the Author):

Sriram Aiyer et al reported overcoming resolution attenuation using tilted cryo-EM data collection. The authors started with highly symmetrical particles with relatively large molecular weight to demonstrate that high-resolution information can be reliably recovered through an optimized data collection and processing strategy. They then showed for relatively “small” particles with D or O symmetry, the high-resolution structure can still be achieved. Mostly importantly, for a dataset showing strong-preferred orientation with C1 symmetry, which represents an authentic scenario where tilted data collection is required, convincingly demonstrate the effectiveness of the data processing pipeline. Furthermore, the authors present a quantitative framework to allows researchers to define an optimal tilt angle for data acquisition. Overall, the work presented here is sound. The methodology and data analysis are well-documented and well-reasoned, which is greatly appreciated by the reviewer.

Response: We thank the reviewer for their positive feedback to our work.

Major question:

The authors first demonstrate that resolution attenuation is minimal for large protein complexes at highly tilted images, which is convincing. However, for many projects which work on relatively small molecules (around 100kDa to 150 KDa), it is generally more difficult to achieve high resolution. The authors have discussed the difficulty for small protein complexes and possible solutions briefly in Line517-524. The question is, will tilted data collection help resolve the preferred-orientation issue for some really challenging samples like some of the membrane proteins in real scenario? Although it has been shown that tilted stage improved density maps for very small proteins at ~43 Kda (Herzik et al Nature Comm 2019), the final resolution is still limited. To strengthen the claim that “resolution attenuation is negligible or significantly across tilt angles”, the authors should demonstrate whether an relatively small molecule (within 200kda, no symmetry) can benefit from the tilted data collection, given the obvious drop in relative SNR of the particles. The drawbacks of tilted data collection should be discussed and highlighted.

*Response: We used three different proteins to demonstrate the ideas reported in the manuscript – AAV (icosahedral), Apoferritin (octahedral), and DPS (tetrahedral). As explained in the manuscript, we needed high-symmetry samples to decouple symmetry-based effects from stage tilt-induced effects on resolution; the samples that we selected *must be* characterized by at least tetrahedral symmetry, but higher-order symmetries are better. The smallest tetrahedral protein that we could find was DPS, which is ~200 kDa.*

We agree with the reviewer that ~200 kDa is not terribly “small” by modern standards, but even this size can present significant challenges, as suggested in Cianfrocco & Kellogg Figure 1¹. As reported in the response to reviewer #2, we also have an unpublished map of a ~120 kDa complex, the structure of which we could only derive using the stage tilt strategy. Because the biology is novel, we cannot report on this structure in the current manuscript, but this project is in the early stages of manuscript preparation and will be reported to the community soon. We

did attempt to reprocess the 43 kDa catalytic domain of protein kinase A dataset from the group of Dr. Gabriel Lander, which is deposited into EMPIAR². Although we observed nominal improvements in resolution, the quality of the map was not clearly improved. We believe that there are other complications with this dataset beyond the small size, possibly due to the accelerating voltage employed for data collection (200 KeV) and the lack of an energy filter. Therefore, we do not wish to report on these results in our current work.

We would also like to draw the attention of the reviewer to two recently published papers that use stage tilt for structure determination of membrane proteins. In the first report, the authors employ up to 30° tilts for the membrane protein OATP1B1 and merge these images with images from untilted and 20° tilted images for their final reconstructions³. OATP1B1 is only 84 kDa in size! In the second report, the authors employ a 45° tilt for ameliorating preferred orientation for the lipoprotein lipase⁴. The lipoprotein lipase is also very small, at ~100 kDa. In both these cases, the final maps yield high-quality atomic models, suggesting that tilted data collection strategies are useful even for membrane proteins whose size is ~100 kDa. Neither of these datasets are publicly available for us to validate the potential improvements in resolution that may be achieved with our processing pipeline. As the reviewer likely knows, membrane proteins pose a lot of additional challenges in addition to adherence to the air/water interface, and thus we believe that further experimental demonstration of our strategies on membrane proteins is beyond the scope of the current work. We now cite these two recent papers in the revised manuscript (Page 16, Line 531).

Collectively, our current work provides a valuable framework for working with tilted data, employing quantitative validation metrics to analyze the maps, and defining optimal tilt angles for data collection. We believe that these tools will be highly useful to the community.

References

¹Cianfrocco, M.A. and Kellogg, E.H., 2020. What could go wrong? A practical guide to single-particle cryo-EM: from biochemistry to atomic models. *Journal of chemical information and modeling*, 60(5), pp.2458-2469.

²Herzik Jr, M.A., Wu, M. and Lander, G.C., 2019. High-resolution structure determination of sub-100 kDa complexes using conventional cryo-EM. *Nature communications*, 10(1), p.1032.

³Ciută, A.D., Nosol, K., Kowal, J., Mukherjee, S., Ramírez, A.S., Stieger, B., Kossiakoff, A.A. and Locher, K.P., 2023. Structure of human drug transporters OATP1B1 and OATP1B3. *Nature Communications*, 14(1), p.5774.

⁴Gunn, K.H. and Neher, S.B., 2023. Structure of dimeric lipoprotein lipase reveals a pore adjacent to the active site. *Nature Communications*, 14(1), p.2569.

Minors

1. For each dataset, a few round of 2D classification were performed to generate equal amount of dataset for comparison, which is methodologically sound. Could the author plot the particle location distribution in the original micrographs? Especially for the high-tilt images, are the particles along the tilt-axis better than particles close to the micrograph edges?

Response: Thank you for the great suggestion. We have now plotted the X and Y coordinates from the AAV2, apoferritin and DPS dataset at different tilt angles, and placed them as **Supplementary Figures 8-10**. Notably, the particle distributions do not aggregate around the tilt axis.

2. Line 323: “Importantly, our results show that with appropriate data processing, tilting the stage is a viable strategy to obtain high-resolution reconstructions even for small proteins suffering from preferred orientation.” DPS dataset does not suffer from preferred orientation issue, therefore this statement is not valid. Also a 230kda complex with tetrahedral symmetry is not a small protein complex.

Response: DPS, like most other proteins, suffers from adherence to the air/water interface, and thus preferred orientation (which can be observed via the Euler plots). However, due to the tetrahedral symmetry and Fourier averaging, the preferred orientation does not lead to an anisotropically resolved reconstruction. This is why we needed minimally tetrahedral symmetry (better, dodecahedral or icosahedral) for our analyses.

We agree with the reviewer that ~200 kDa is not particularly small, but there are now other examples of smaller proteins whose structures have been determined using tilts. Please see our detailed response to the critique above. We have amended the statement as follows, “Importantly, our results show that with appropriate data processing, tilting the stage can be a viable strategy to obtain high-resolution reconstructions even for relatively small proteins suffering from preferred orientation” (Page 10, Lines 325-327).

In line with the major critique above, we amended the original statement in the discussion:

“Nevertheless, stage tilt has been employed for proteins as small as ~43 kDa to improve map quality¹”

to cite the recent papers and now read (Page 16, Lines 530-531):

“Nevertheless, stage tilt has been employed for proteins as small as ~43 kDa to improve map quality¹, including membrane proteins in the ~100 kDa range^{2,3}”

References

¹Herzik Jr, M.A., Wu, M. and Lander, G.C., 2019. High-resolution structure determination of sub-100 kDa complexes using conventional cryo-EM. *Nature communications*, 10(1), p.1032.

²Ciută, A.D., Nosol, K., Kowal, J., Mukherjee, S., Ramírez, A.S., Stieger, B., Kossiakoff, A.A. and Locher, K.P., 2023. Structure of human drug transporters OATP1B1 and OATP1B3. *Nature Communications*, 14(1), p.5774.

³Gunn, K.H. and Neher, S.B., 2023. Structure of dimeric lipoprotein lipase reveals a pore adjacent to the active site. *Nature Communications*, 14(1), p.2569.

3. L543-544, The actual ice layer tilt angle could be slightly different to the tilt angle. Could the authors provide the calculated tilt angle distribution for each tilted dataset? This is provided from cryoSPARC.

Response: This is a great idea, and we have incorporated this information into all of our supplementary tables.

4. Are the microscopes used in this study single-tilt stage or double-tilt stage?

Response: We used single-tilt stage for all of the datasets collected at Scripps and Janelia Research Campus. The proteasome data was collected on a dual-tilt stage at NYSBC.

5. In general, a relatively small dataset was collected from each sample. For example, in Line910-911, “A total of 637 micrographs was recorded over a single 4-day collection using stage shift, across tilt angles of 0° and 30° set with stage alpha.” This is considerably lower throughput compared with modern capacity of Krios microscope. Could the author provide more information regarding the speed of data collection? Or is a longer waiting time is setup for stage drift? Any difference in terms of stage settling time at different stage tilt? These information will be useful for readers to transfer the knowledge of this study to other applications.

Response: The reviewer is correct – our throughput was low. This is because the data collections were not actually performed continuously. We did not collect overnight, and we were careful about having identical conditions for accurate comparisons between different stage tilt angles for this work, specifically.

However, in our experience with other biological samples in the lab, we simply set the stage tilt angle and collect as usual, and even overnight. The throughput is similar to the throughput for collecting untilted images. For clarity, we have now added the following statement to the Methods section (Page 33, Lines 905-907; Page 34, Lines 908-910).

“We note that the AAV2 data collection, and all other subsequent data collections, were not performed continuously, because we were concerned with having identical conditions for accurate comparisons between different stage tilt angles. Modern workflows for data collection can yield higher throughput, especially using beam-tilt induced image-shift and active beam-tilt compensation for targeting. Once the tilt angle is defined and the imaging conditions are established, the throughput for collecting data using stage tilt should rival that of collecting data without stage tilt.”

REVIEWERS' COMMENTS

Reviewer #1 (Remarks to the Author):

The comments of this reviewer are satisfied - no further suggestions. Publication recommended.

Reviewer #2 (Remarks to the Author):

The Authors have addressed and clarified points raised in the earlier revision. I look forward to seeing it published, and availability of the raw data on EMPIAR.

Reviewer #3 (Remarks to the Author):

The authors have addressed all of my comments and I do not have further comments.
Very solid comparative study for tilted data collection strategy.